# Genomic signatures associated with maintenance of genome stability and venom turnover in two parasitoid wasps

Xinhai Ye [1,2,3,7], Yi Yang [1,7], Can Zhao [4,7], Shan Xiao [1], Yu H. Sun [5], Chun He[1], Shijiao Xiong [1], Xianxin Zhao[1], Bo Zhang[1], Haiwei Lin[1], Jiamin Shi[1], Yang Mei [1], Hongxing Xu[6], Qi Fang[1], Fei Wu[2,3], Dunsong Li [4] ✉ & Gongyin Ye [1] ✉

Parasitoid wasps are rapidly developing as a model for evolutionary biology. Here we present chromosomal genomes of two *Anastatus* wasps, *A. japonicus* and *A. fulloi*, and leverage these genomes to study two fundamental questions—genome size evolution and venom evolution. *Anastatus* shows a much larger genome than is known among other wasps, with unexpectedly recent bursts of LTR retrotransposons. Importantly, several genomic innovations, including *Piwi* gene family expansion, ubiquitous *Piwi* expression profiles, as well as transposable element-piRNA coevolution, have likely emerged for transposable element silencing to maintain genomic stability. Additionally, we show that the co-option evolution arose by expression shifts in the venom gland plays a dominant role in venom turnover. We also highlight the potential importance of non-venom genes that are coexpressed with venom genes during venom evolution. Our findings greatly advance the current understanding of genome size evolution and venom evolution, and these genomic resources will facilitate comparative genomics studies of insects in the future.

Parasitoid wasps comprise extremely diverse species[1,2]. Accumulating genomic data and their unique characteristics have made parasitoid wasps become central models for understanding fundamental questions in evolutionary biology, such as genome size evolution and venom evolution. The eukaryotic lineages display a broad diversity of genome sizes; however, the evolutionary forces and molecular mechanisms behind this have puzzled scientists over the decades[3–5]. Comparative genomics analyses involving organisms with a wide spectrum of genome sizes indicated that transposable elements (TEs) are one of the key factors driving genome size evolution[6–9]. During the TE-driven genome expansion, the increased TE activity may inevitably cause genomic instability, potentially resulting in overloaded double-stranded breaks and gene disruptions causing cell death[10]. This raises a central question in genome evolution: how does the genome evolve to control the sudden bursts of TE activity? Recently, studies of ultra-large genomes focusing on the African lungfish[11] and the Chinese pine[12]

[1]State Key Laboratory of Rice Biology & Ministry of Agricultural and Rural Affairs Key Laboratory of Molecular Biology of Crop Pathogens and Insects, Institute of Insect Sciences, Zhejiang University, Hangzhou, China. [2]Shanghai Institute for Advanced Study, Zhejiang University, Shanghai, China. [3]College of Computer Science and Technology, Zhejiang University, Hangzhou, China. [4]Institute of Plant Protection, Guangdong Academy of Agricultural Sciences, Key Laboratory of Green Prevention and Control on Fruits and Vegetables in South China Ministry of Agriculture and Rural Affairs, Guangdong Provincial Key Laboratory of High Technology for Plant Protection, Guangzhou, China. [5]Department of Biology, University of Rochester, Rochester, NY, USA. [6]State Key Laboratory for Managing Biotic and Chemical Treats to the Quality and Safety of Agroproducts, Institute of Plant Protection and Microbiology, Zhejiang Academy of Agricultural Sciences, Hangzhou, China. [7]These authors contributed equally: Xinhai Ye, Yi Yang, Can Zhao. ✉e-mail: dsli@gdppri.cn; chu@zju.edu.cn

showed impressive genomic innovations in TE control, including KRAB domain expansion and widespread DNA methylation, likely reflecting the adaptive evolution of genomic arms race with TEs. In insects, although TE-induced genome expansions have been documented in some lineages[9, 13], the genomic and molecular evolutionary features of TE control are scarce. Generally, two main strategies empower TE control in insects: PIWI-interacting RNA (piRNA)[14] and DNA methylation[15]. piRNAs mainly protect the germline by base-pair complementarity targeting TEs[16–18], while this mechanism can be extended beyond the germline in some insects such as aphids, honey bees, beetles and mosquitos[14, 19]. Additionally, DNA methylation is another TE suppressor in some hemimetabolous insect lineages, including locust, mealybug, and cockroach, but this strategy is largely lost in holometabolous insects, including the parasitoid wasp *Nasonia vitripennis*[15, 20, 21]. With recently assembled large genomes supporting the extensive amplification of TEs[22], parasitoid wasps have become promising subjects for studying insect genome evolution.

Parasitoid wasps use their venom as a major weapon against their hosts[23]. To adapt to the varied hosts, parasitoid venom repertoires evolve rapidly, providing an excellent model for understanding how genes evolve to acquire their new "job" in venom[24]. A previous study examined the venom genes of four parasitoid wasps, and proposed that the co-option of single-copy genes through expression alternations may be a common mechanism of their venom evolution[24]. However, the applicability to other parasitoid venom systems and the genetic basis of this mechanism require further investigations. Moreover, the gene regulatory networks associated with venom genes, and their evolutionary trajectories during venom evolution remain poorly understood, limiting our knowledge of parasitoid venom evolution.

Here we report two ~950 Mb chromosomal genomes of *Anastatus japonicus* and *A. fulloi*, two important parasitoid wasps that have been widely used for pest control in China for over a half-century[25] (Supplementary Fig. 1). With these high-quality genomes and additional omics data, we show that their genome sizes are mainly enlarged due to recent bursts of long terminal repeat (LTR) retrotransposons, which presumably occurred prior to their divergence. In adaptive response, their genomes strengthen the ability of TE silencing through expanding the *Piwi* gene family, widely expressing them across development and throughout the whole body, and coevolving with the piRNA dynamics. We also discover that, in *Anastatus* wasps, co-option evolution arose by expression shifts in the venom gland plays a predominant role in rapid venom turnover. In addition, we construct the venom-related gene regulatory networks and found many non-venom genes coexpressed with venom genes. Comparative analyses further suggest the potential importance of these non-venom genes in venom gene regulation and venom evolution.

## Results

### Genomic features of two *Anastatus* wasps, *A. japonicus* and *A. fulloi*

We employed PacBio high-fidelity (HiFi) long-read sequencing and Illumina short-read sequencing technologies to generate high-quality contigs for two *Anastatus* wasps, *A. japonicus* and *A. fulloi* (Supplementary Tables 1 and 2). These contigs were further scaffolded using Hi-C libraries to yield chromosome-level genome assemblies comprising five chromosomes for each wasp (Supplementary Fig. 2; Supplementary Tables 3 and 4), consistent with the haploid number of these wasps ($n = 5$) assessed by chromosome staining experiment[26]. The final assembled genome size of *A. japonicus* and *A. fulloi* is 950.9 and 963.4 Mb, respectively, with long scaffold N50 lengths (178.3 and 177.0 Mb) (Fig. 1a, Supplementary Table 5), which is approximately equal to the estimated genome size by *k*-mer analysis (Supplementary Fig. 3, Supplementary Table 6). The assemblies are of high integrity and accuracy as over 97% of complete BUSCO genes have been identified and over 99% of Illumina reads can be successfully mapped back

to each wasp assembly (Supplementary Tables 7 and 8). A total of 27,792 and 27,168 protein-coding genes were annotated in *A. japonicus* and *A. fulloi*, respectively, and both of them were supported by over 94% BUSCO completeness, suggesting the comprehensive gene annotation (Supplementary Table 7). To further validate the quality of genome assembly, we generated ultra-long DNA sequencing reads for each species using the Oxford Nanopore Technologies (ONT) platform, and mapped them back to the genomes. We found that, in both wasps, more than 99% of these reads aligned to assembly scaffolds, including over 30,000 reads longer than 100 Kb that mapped uniquely and consistently (Supplementary Tables 9 and 10). Mapped sequencing read depth showed uniform coverage across all chromosomes in both wasps, with 99.9% of the assembly having coverages within three standard deviations of the mean values for either PacBio HiFi or ONT (Supplementary Figs. 4–6, Supplementary Table 11). We also observed the uniform sequencing coverage between the paralogs of duplicated BUSCO genes in the chromosome-scale scaffolds of these two wasps (Supplementary Figs. 7–55). Overall, these results confirmed the accuracy of the assemblies. By comparing the two chromosome-level genomes, we revealed an overall high synteny conservation between *A. japonicus* and *A. fulloi* using whole-genome alignment analysis (Fig. 1a), suggesting few large-scale genome rearrangements occurred after speciation, although 2070 structural variations (>10 Kb) were still identified by an assembly-based detection[27] (Supplementary Data 1).

The most prominent feature of these two *Anastatus* genomes is that they are among the largest parasitoid wasp genomes sequenced to date, accounting for 3–4 times of most parasitoid wasp genomes accessible[22, 28]. We found massive amounts of TEs (about 59%) and other interspersed repeats in their genomes (Supplementary Data 2). Moreover, both genomes contain a substantial number of intact TEs that are probably active (Supplementary Data 3 and 4). Thus, these unique genomic characteristics make them a promising model for investigating several fascinating questions about genome size and TEs, such as the co-evolution of TE propagation and genome-wide repression strategies, and how TE insertions shape gene expression, which will be highlighted below.

### Phylogenomics and gene family evolution

We demonstrated the phylogenetic relationships among the two *Anastatus* wasps and 17 other representative hymenopteran species by constructing a maximum-likelihood phylogenetic tree with 1792 one-to-one orthologous proteins (Fig. 1b). In our analysis, *Anastatus* wasps were placed as the sister group to Pteromalidae species (*Nasonia* and *Pteromalus*), and their divergence was estimated at around 78 million years ago (mya). Additionally, we also found that these two *Anastatus* species diverged approximately 3 mya. The topology of this phylogenetic tree is consistent with prior studies based on transcriptomic data[29].

We next analyzed the gene family expansions and contractions and discovered a significant number of gene family expansions in both *Anastatus* wasp lineages, together with their common ancestor, compared to the relatively older nodes (Fig. 1b). This phenomenon is likely related to another noticeable aspect of these two *Anastatus* genomes: they encode more genes than other parasitoid wasps[22, 30, 31], and they have high gene gain rates compared to the other species in our analysis (434 and 317 per My for *A. japonicus* and *A. fulloi*, Supplementary Fig. 56, Supplementary Data 5). We also found relatively high gene loss rates of 51 and 56 per My for the terminal branches of these two wasps, respectively. Here, we only focused on the gene family evolution that occurred in the common ancestor of *Anastatus* and sought to decipher the causal connections between the shared genomic changes and phenotypic novelties in *Anastatus* wasps. In the common ancestor of *Anastatus*, 2094 gene families were expanded, enriching genes associated with muscle stretch, piRNA metabolic pathways, and Toll-like

receptor signaling pathways (FDR-adjusted $p < 0.05$, Supplementary Data 6). The expanded gene families involved in muscle stretch may be associated with their newly evolved ability to jump[32]. Notably, the gene family expansion related to piRNA metabolic pathways is likely a responsive strategy for *Anastatus* wasps to combat the largely expanded TEs in their genomes. Moreover, we found a larger number of olfactory receptor genes in both *Anastatus* (104 and 102) and *Nasonia* (72) genomes compared to honey bee (22), with a highly duplicated 9-exon subfamily. These duplicated genes in *Anastatus* may be associated with their wide host ranges of at least 16 recorded hosts across seven families, falling into two orders[33] (Supplementary Fig. 57 and Supplementary Table 12). Additional enrichment results about the gene family evolution in these two *Anastatus* wasp lineages can be found in Supplementary Data 7 and 8.

## Recent bursts of LTR retrotransposons drive genome size evolution in *Anastatus*

According to the sequencing projects completed so far, the majority of hymenopteran genomes are smaller than 500 Mb (Supplementary Data 9), with only a few exceptions, such as the pincer wasp *Gonatopus flavifemur* (636 Mb)[22]. The sequencing of these two *Anastatus* genomes (about 950 Mb) provides additional examples for us to delve deeper into the mechanisms underlying genome enlargement in Hymenoptera.

Nearly 60% of each *Anastatus* genome consists of repetitive elements (over 98.5% of repetitive elements are TEs) that are distinctly enriched in the central region of each chromosome, where centromeres may locate. On the contrary, coding genes distribute toward both ends of each chromosome (Fig. 1a). The most abundant TEs in the

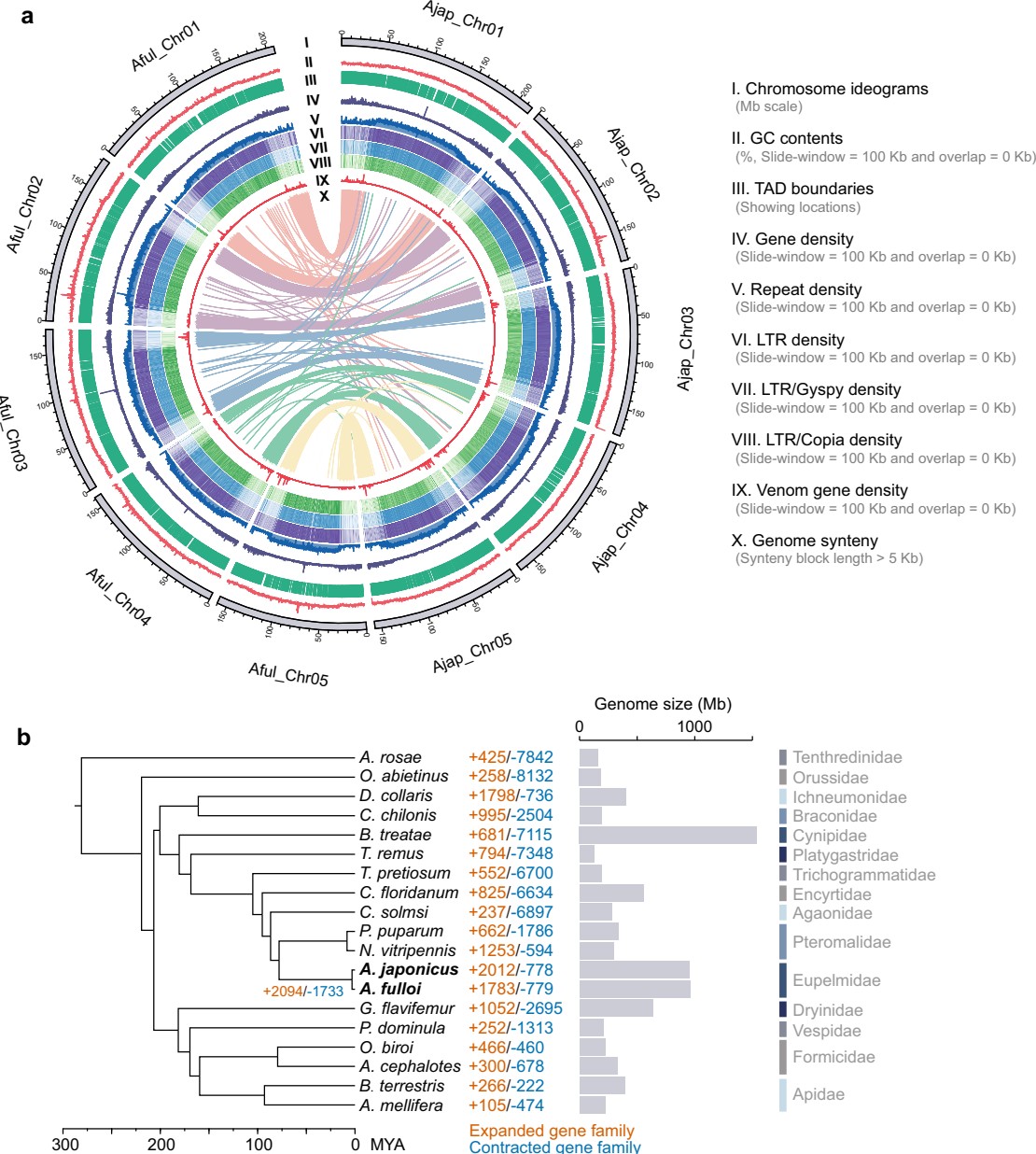

**Fig. 1 | Genomic architecture of the two *Anastatus* genomes and hymenopteran phylogeny. a** Genomic landscapes of *A. japonicus* and *A. fulloi*. The denotation of each track is listed on the right of the circos plot. **b** A maximum-likelihood phylogenetic tree constructed using concatenated protein sequences of 1792 one-to-one orthologous genes from 19 hymenopteran species and 9 calibration points. 1000 bootstraps suggest that all nodes have 100% support. The gene family numbers of expansion (in orange) and contraction (in blue) on terminal branches and common ancestor branch of two *Anastatus* wasps are indicated. The bar plot adjacent to the phylogenetic tree shows the genome sizes of insects in this study. Source data are provided as a Source Data file.

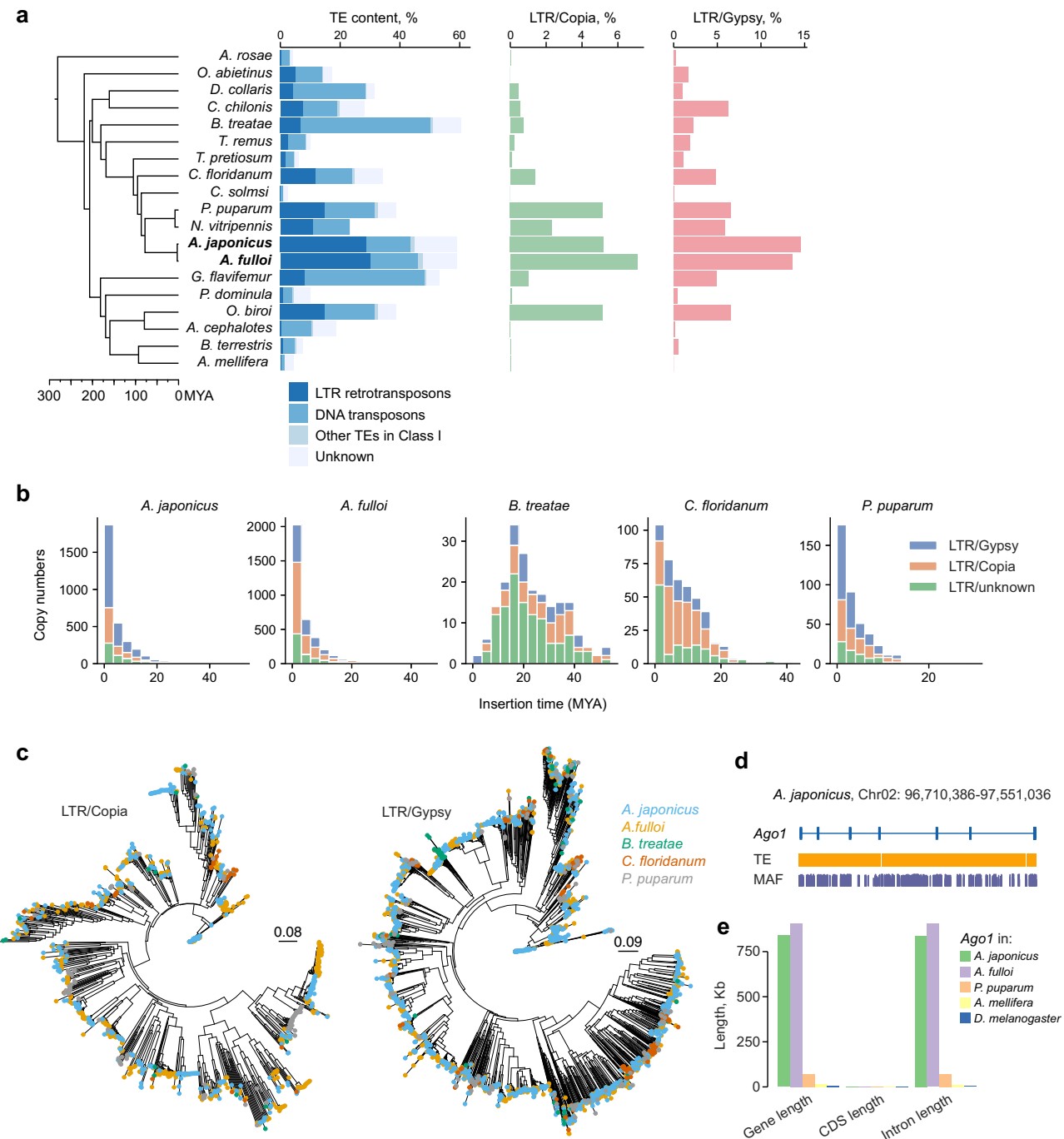

**Fig. 2 | Genome size enlargement and TE expansion in the *Anastatus* genomes.**
**a** Comparison of TE contents in various hymenopteran species. The schematic tree on the left shows evolutionary relationships and is obtained from Fig. 1c. **b** Estimated insertion time of full-length LTR retrotransposons in five hymenopteran insects. **c** Neighbor-joining trees constructed using 1439 full-length *Copia* and 3140 *Gypsy* sequences from five hymenopteran insects. **d** Massive amounts of TE insertions in the introns of *Ago1* in the *Anastatus* genomes, which resulted in a birth of a long gene. This plot is generated by the UCSC Genome Browser for *A. japonicus*. The TE track indicates the distributions of TEs and the MAF track represents the whole genome alignment between *A. japonicus* and *A. fulloi*. **e** Bar plots comparing the gene length, CDS length, and intron length of *Ago1* among five hymenopteran insects and one *Drosophila* species. Source data are provided as a Source Data file.

two *Anastatus* genomes are LTR retrotransposons, which comprise 28.7% and 30.0% of *A. japonicus* and *A. fulloi* genome, respectively (Fig. 2a; Supplementary Data 2). However, DNA transposons are predominant TEs in most other hymenopteran genomes we examined. Principal component analysis of TE composition further confirmed that the TE landscape (proportions of major classes of TEs) of *Anastatus* is unique among hymenopterans (Supplementary Fig. 58a).

Within LTR retrotransposons, the most abundant elements in the two *Anastatus* genomes are *Gypsy*-type LTRs (about 13–14% of the genomes, and about 130 Mb in length) (Fig. 2a; Supplementary Data 10). And the *Copia*-type LTR retrotransposons make up about 5–7% of the genomes. Although the proportions of *Copia* elements in *Anastatus* appear to be comparable to some other hymenopterans, the total length of *Copia* LTRs (mean of their total length is 59 Mb) is much longer than any other hymenopterans (mean of their total length is 4 Mb). Additionally, we noted that approximately 9% of the genomes are classified as unknown LTR retrotransposons, which may contribute to genome evolution as well. We also found a slight difference in the

LTR retrotransposon content between these two *Anastatus* wasps, suggesting that some small-scale LTR retrotransposon expansions or eliminations occurred after the species divergence. To further investigate the expansion pattern of LTR in these two *Anastatus* genomes, we firstly identified the full-length LTR retrotransposons, and estimated their insertion time by calculating the sequence divergence between adjacent 5´ and 3´ LTRs of the same retrotransposon (Supplementary Data 3 and 4). In both two *Anastatus* genomes, the results indicated a clear burst of LTR retrotransposon activity within the last 1–3 mya (Fig. 2b). We inferred that they shared the same LTR retrotransposon burst event, which happened in the common ancestor of the two *Anastatus* species, based on their highly similar sequence divergence pattern (Supplementary Fig. 58b). However, we cannot rule out the possibility that independent LTR retrotransposon bursts occurred after their divergence, because the estimated species divergence time is very close to the estimated time of LTR retrotransposon expansion. Phylogenetic analysis of reverse transcriptase genes from full-length LTR retrotransposons (*Gypsy* elements and *Copia* elements) revealed several large *Anastatus*-expansion clades and species-specific clades, which also corroborates our finding on the recent bursts of LTR retrotransposons in the two *Anastatus* species (Fig. 2c).

We also hypothesized that, in addition to the large LTR retrotransposon expansions, these *Anastatus* genomes have lower efficiency in eliminating TE compared to other species. The ratio of solo-LTR and paired-LTR (intact LTR) was used to test the hypothesis, as the solo-LTRs are mainly caused by DNA removal via unequal homologous recombination[34]. However, the results showed that the ratios of the two *Anastatus* (29 and 25) were comparable to those of most other wasps. Thus, we failed to uncover strong evidence for our hypothesis proposed above.

## Potential impacts of TE insertions in *Anastatus* genomes

We next investigated the influence of TE on gene structure in the two *Anastatus* genomes. The overall gene structure patterns of the two *Anastatus* species, including the length of gene, coding sequence (CDS), exon, and intron, are similar to a closely related species *P. puparum*, but differ from those of the distant species (Supplementary Fig. 59a–d). Additional comparisons using 1792 strict one-to-one orthologous genes among 19 hymenopterans also showed that the median CDS length and intron length are highly similar in the *Anastatus* and other species, although we observed species-specific enlargement of introns in *Belonocnema*, another wasp with a large genome size (1.5 Gb) (Supplementary Fig. 59e, f).

Nevertheless, we found a few cases (23) in *Anastatus* where the expansion of TE in introns eventually led to the emergence of extra-long genes ($p < 0.05$, two-sided Wilcoxon rank-sum test; Supplementary Data 11). For instance, Argonaute 1 (*Ago1*), a key gene in miRNA pathways, had undergone clear intron extension by TE insertions in both *Anastatus* wasps, resulting in an extra-long gene of over 840 Kb in each species, whereas orthologous genes in other hymenopterans (72 Kb for *P. puparum* and 14 Kb for *A. mellifera*) and *Drosophila* (6.8 Kb) are much shorter (Fig. 2d, e). Whole-genome alignment analysis between *Anastatus* wasps revealed a number of introns with low conservation, suggesting continuous TE insertion events following speciation (Fig. 2d). Although considerable changes have occurred in *Anastatus Ago1*, the entire transcript could be successfully detected in their transcriptomes with reasonable expression levels (TPM = 40–100), indicating that changes in *Ago1* length may not impair its gene expression in *Anastatus*.

To investigate whether recent TE activities had influenced gene expression patterns after species divergence, we measured the consistency of gene expression across development between the orthologous gene pairs in two *Anastatus*, and tested if the expression correlation coefficient decreased when recent TE insertion happened explicitly in the potential regulatory regions (1 Kb upstream or downstream) in one species. Our result revealed a substantial decrease in the expression correlation coefficient between gene pairs, one of which has a recent TE insertion, but the other does not ($p = 1.96e−12$, two-sided Wilcoxon rank-sum test; Supplementary Fig. 59g).

## Uniquely massive expansion of the *Piwi* gene family

Our enrichment analysis on the expanded gene families of *Anastatus* drew our attention to *Piwi* genes (from the Argonaute superfamily) and piRNAs, which play vital roles in suppressing TEs[35]. In the *Anastatus* species, we observed a substantially larger repertoire of *Piwi* genes than all other hymenopterans or outgroups (beetle and fly). Our bioinformatic search (using tblastn/blastp, Fgenesh+, and HMMER) and manual curation identified 30 and 16 *Piwi* genes in the chromosome-level scaffolds of *A. japonicus* and *A. fulloi*, respectively; however, other species we studied have much fewer *Piwi* genes (zero to six) (Fig. 3a). We confirmed that the presence of a large number of *Piwi* genes was not a consequence of genome assembly errors by inspecting PacBio HiFi and ONT reads, which display uniform read coverage across the whole regions (Supplementary Figs. 60–76, Supplementary Tables 13–16, Supplementary Data 12 and 13). In both wasps, we identified tandem-duplicated *Piwi* genes forming *Piwi* clusters, which can be supported by uniquely mapped ONT ultra-long reads (Supplementary Fig. 77). In contrast, the gene counts of other Argonaute superfamily members (*Ago1*, *Ago2*, and *Ago3*) are relatively consistent across Hymenoptera (Fig. 3a). We also found numerous sequences of *Piwi* on some short contigs with few other genes, suggesting that challenges remain to assemble them into their chromosome-level scaffolds due to their repetitive gene structures, or, they could be redundant artifacts. Given this uncertain status, we only include the *Piwi* genes that are confidently assigned to chromosomes in the downstream analysis.

Next, we constructed a phylogenetic tree of all Argonaute superfamily members from nine hymenopterans and model insect *D. melanogaster* to understand the dynamic evolution of these genes (Fig. 3b). The *Piwi* tree can be divided into two main clades—Hymenoptera (Hym)-conserved clade (containing the genes from all hymenopterans) and Chalcidoidea (Cha)-specific clade, supporting the existence of an ancient duplication prior to the Chalcidoidea divergence. Evolutionary rate analysis revealed an asymmetrical pattern after this duplication, with Cha-specific clade showing a higher level of divergence than Hym-conserved clade ($dN/dS_{\text{Cha-specific clade}} = 0.115$, $dN/dS_{\text{Hym-conserved clade}} = 0.05$, $p < 0.0001$; Supplementary Table 17), probably implying the functional importance of the Cha-specific clade. There are numerous well-supported species-specific clades within the Cha-specific clade (mainly in *A. japonicus* and *A. fulloi*), consistent with repeated duplications of *Piwi* genes following the divergence of *A. japonicus* and *A. fulloi*. In addition, six clear sub-clades in the Cha-specific clade containing both *Anastatus* species further indicated several duplications occurred in the common ancestor of *Anastatus*.

To examine the *Piwi* expression pattern in the two *Anastatus* species, we carried out RNA-seq analysis and found a broad expression pattern of these *Piwi* genes across almost all developmental stages (Supplementary Fig. 78c, d). Moreover, quantitative reverse transcription PCR (qRT-PCR) analyses showed varied expression patterns of these *Piwi* genes in different adult wasp tissues (Fig. 3c, Supplementary Fig. 78a, b). Specifically, we measured *Piwi* expression on mRNA level in the reproductive organs (ovary and testis) and other somatic tissues. In *A. japonicus*, as expected, we observed the highest expression of all *Piwi* genes in the reproductive organs. However, in addition to finding *Piwi* genes that were specifically highly expressed in reproductive organs (e.g., *Piwi6* in Fig. 3c), we also found genes with significantly detectable expressions in the other analyzed tissues as well. For example, *Piwi2* was also highly expressed in the head compared to other tissues ($p < 0.05$, Tukey's multiple comparison test), reaching 45% of testis expression. Also, in relation to testis expression,

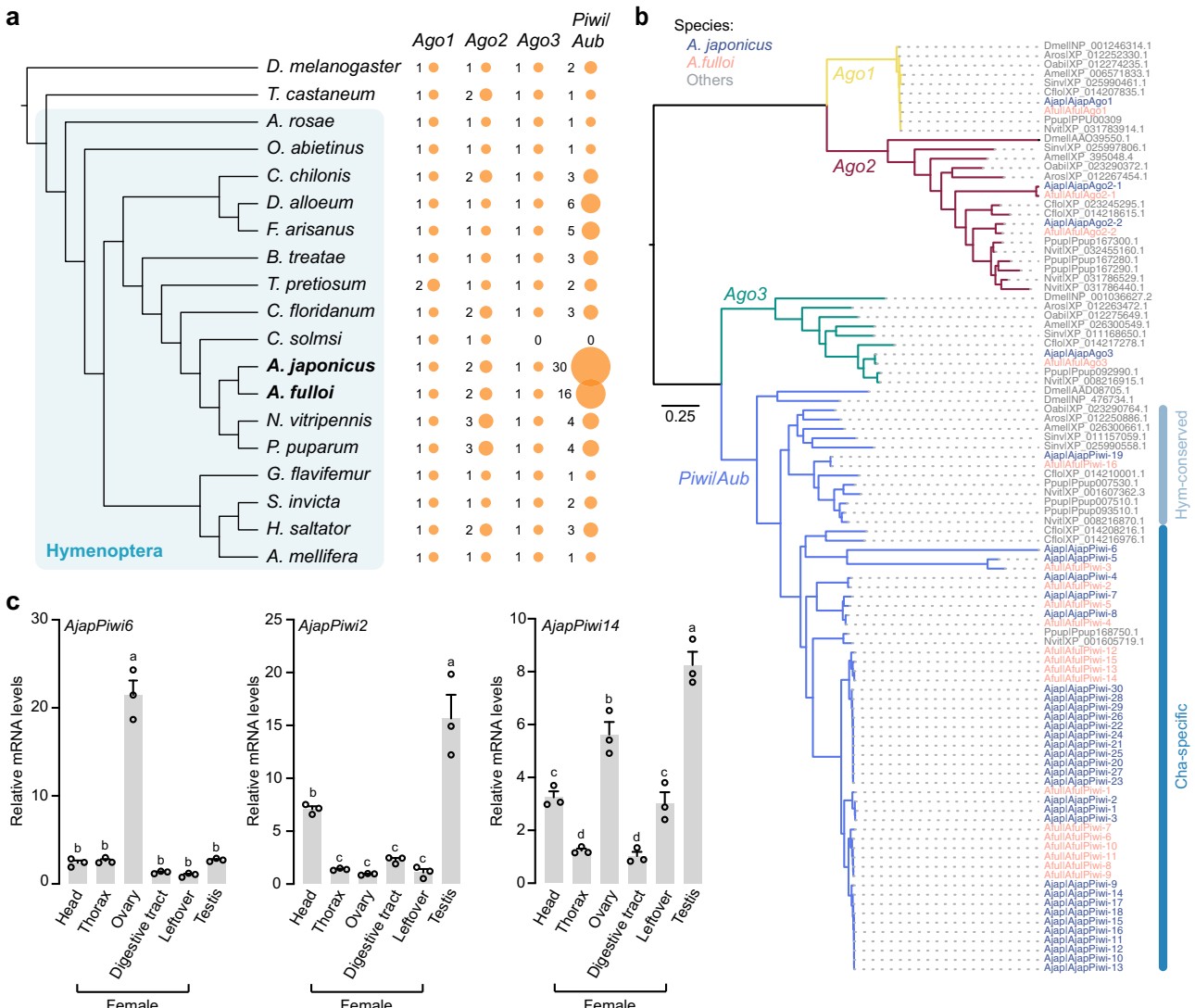

**Fig. 3 | Massive *Piwi* expansions in the *Anastatus* genomes. a** Key members of argonaute superfamily across 17 hymenopterans, *Tribolium castaneum* (Coleoptera) and *Drosophila melanogaster* (Diptera). The sizes of the orange bubbles represent the number of genes of a particular family. **b** Phylogenetic tree of the genes from the Argonaute superfamily in nine hymenopterans and *D. melanogaster*. CDS sequences were used for the analysis, and IQ-TREE was applied for tree construction with 1000 ultrafast bootstrap replicates. **c** qRT-PCR based gene expression of three representative *Piwi* genes (*Piwi6*, *Piwi2* and *Piwi14*) in the different tissues of adult *A. japonicus* wasp. The head, thorax, ovary, digestive tract and leftover from female adult wasps and testis from male adult wasps are used for analysis. Data represent means ± SEM from three biological replicates. Lowercase letters indicate significant differences (one-way ANOVA followed by Tukey's multiple comparison test, *p* < 0.05). Exact *p* values are presented in Supplementary Data 33. Source data are provided as a Source Data file.

*Piwi14* was expressed in diverse levels ranging from 12% (in digestive tract) to 68% (in ovary). Our results therefore imply that, in the *Anastatus* species, numerous *Piwi* genes may function with piRNAs coordinately to defend TEs with temporal and spatial dynamics, consistent with the finding that the piRNA pathway is not restricted to the germline in many arthropods, including hymenopteran species[14]. Together, we hypothesized that, along with the recent TE bursts, massive *Piwi* gene duplication and extensive expression of these *Piwi* genes have evolved in the *Anastatus* species, enhancing their ability to repress TE activity via the piRNA pathway, representing a paradigmatic example of the arms race between TEs and the host genome.

**piRNA profiles and their functions in silencing TEs**
We next performed small RNA sequencing to further dissect the features and functions of piRNAs in the two *Anastatus* genomes. Specifically, we aimed to address three questions: (1) Do these two wasps produce piRNAs? (2) If so, do their piRNAs control TEs in the

genomes? and (3) If so, what's the relationship between piRNA profiles and TE characteristics (i. e., co-evolution between piRNAs and TEs)? Because of the widespread presence of somatic piRNAs in arthropods including hymenopterans[14], whole female adults, rather than just germline, were used for sequencing. As a result, our bioinformatic analysis captured a bimodal length distribution of small RNAs peaking at 21–23 nt and 27–29 nt in both *Anastatus* species (Fig. 4a, b; Supplementary Tables 18 and 19). The first peak was dominated by miRNAs and siRNAs (Supplementary Fig. 79). Reads from the second peak exhibited a strong "U" bias at the first nucleotide (1U), and a significant proportion of them also had an "A" bias at the 10th position (1U & 10A), a signature of piRNAs generated by the Ping-Pong amplification[36]. Therefore, these profiling results suggest that abundant piRNAs are present in both *Anastatus* wasps. We further filtered out reads mapped to annotated non-coding RNAs and obtained a clean dataset containing small RNAs between 24–35 nt, the typical piRNA length.

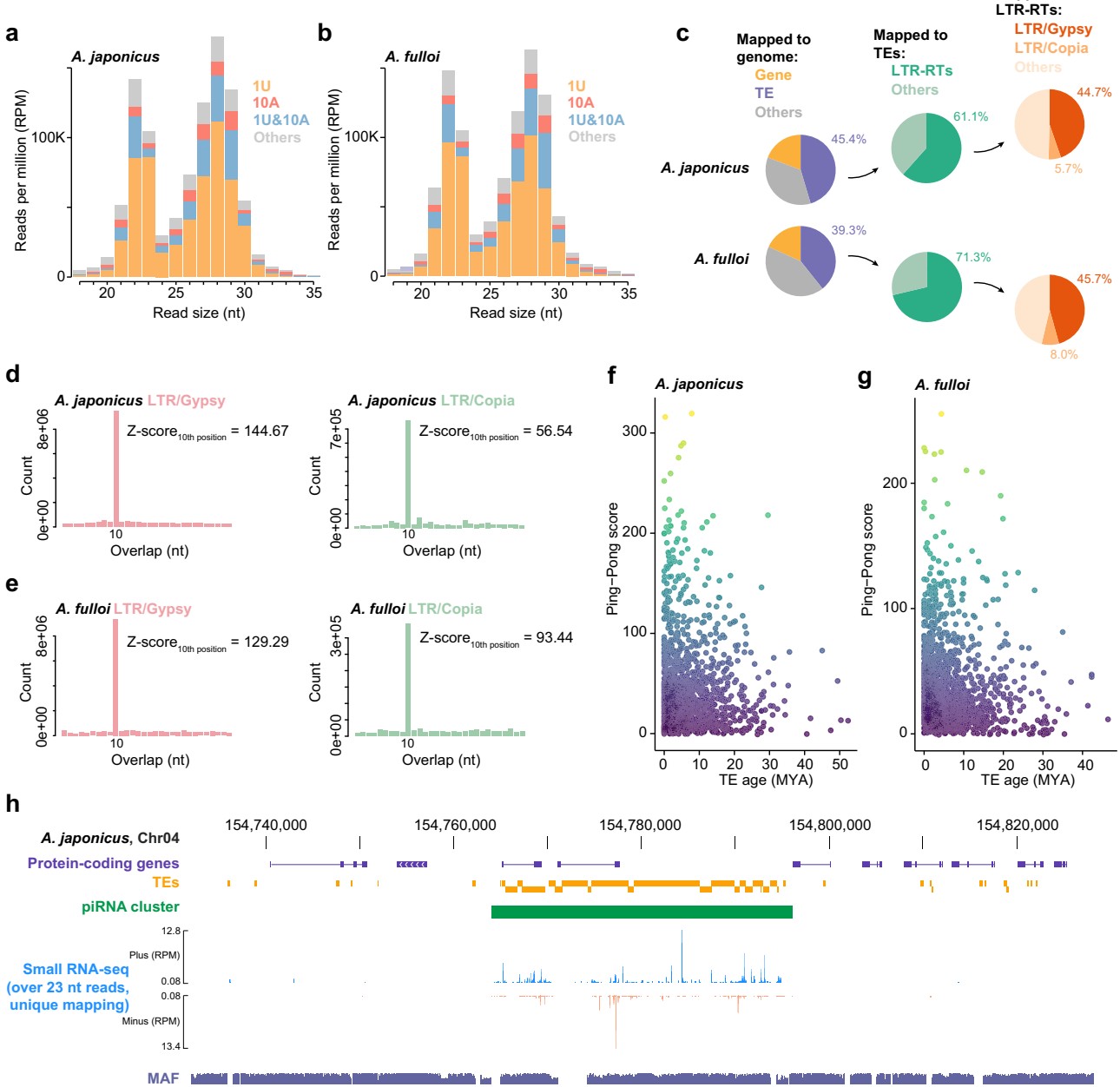

**Fig. 4 | piRNAs in the *Anastatus* genomes.** Distribution of small RNA read sizes mapping to the *A. japonicus* genome (**a**) and the *A. fulloi* genome (**b**). Four kinds of reads are shown in different colors in each bar in the size distribution. **c** Mapping statistics of piRNA reads to the two *Anastatus* genomes. **d** Analysis of the 5′-to-5′ distance between piRNAs mapped to opposite strands of two types of LTR retrotransposons (*Gypsy* and *Copia*) in *A. japonicus*. The same analysis of *A. fulloi* is also shown (**e**). The significance of ten-nucleotide overlap ('Ping-Pong') was determined using Z-score. **f**, **g** Scatter plots illustrating the relationship between TE ages (LTR retrotransposons) and Ping-Pong scores in the two *Anastatus* genomes. **h** Genome browser view revealing a large number of piRNAs produced from a TE-rich region in the *A. japonicus* genome. Source data are provided as a Source Data file.

We found about 45.4% and 39.3% of total piRNAs mapped to TEs in the *A. japonicus* genome and *A. fulloi* genome, respectively (Fig. 4c). These statistics were much higher than those (<20%) of other reported hymenopterans (bees)[14]. Genome browser view confirmed clustered piRNA reads mapped to a TE-rich region, providing a representative example of the piRNA clusters (totally 1111 and 1020 for *A. japonicus* and *A. fulloi*, respectively) against TEs (Fig. 4h, Supplementary Fig. 80, Supplementary Data 14, 15). Among TE-mapping piRNAs, the majority of them were LTR-RT(LTR retrotransposon)-targeting piRNAs (61.1% and 71.3% in *A. japonicus* and *A. fulloi*), consistent with the TE features (i. e., LTR-RT rich) of the *Anastatus* genomes. Within LTR-RT-targeting piRNAs, 44.7% and 45.7% of them are mapped to *Gypsy* elements in *A.*

*japonicus* and *A. fulloi*, respectively. We analyzed the 5′-to-5′ distance between piRNA pairs mapped to opposite strands of TEs, and observed a significant Z score of 10 nt in the piRNAs derived from different TE types (for example, Fig. 4d, e; Supplementary Fig. 79g–j), indicating the existence of Ping-Pong amplifications. In addition, the Ping-Pong scores of LTR retrotransposons were more pronounced compared to the other TEs, suggesting higher effectiveness of piRNA pathway against LTR retrotransposons, which potentially antagonized the recent bursts of LTR retrotransposons in *Anastatus*. We next investigated how TE ages are associated with Ping-Pong scores (i. e., piRNA activity). We hypothesized that the Ping-Pong amplification is stronger in young TEs than the old ones, as the young TEs are thought to be

more active[37]. In the two *Anastatus* genomes, old LTR retrotransposons showed relatively lower Ping-Pong scores, while the Ping-Pong scores of young LTR retrotransposons were variable (Fig. 4f, g).

Together, we demonstrated that abundant piRNAs are present in the two *Anastatus* genomes, and a substantial number of them are involved in TE silencing. Moreover, we found that the metrics of TE-derived piRNAs, including piRNA content and Ping-Pong signals, mirror those of TE in the genomes, implying an adaptive response of piRNAs to TE evolution.

## Rapid venom evolution in *Anastatus*

Rapid venom turnover is one of the most remarkable features in parasitoid wasp evolution, and it is of great interest to explore how genes with new functions arise[24]. Venoms are produced in the venom gland of adult female parasitoid wasps, and then transferred and stored in a downstream organ called venom reservoir until being used. Genes in the venom gland showed a distinct expression pattern compared to other developmental stages and tissues (Fig. 5a), since only a small set of genes expressed at high levels, i.e., roughly 2% of all genes account for 90% of transcriptomic reads in the venom gland (Supplementary Fig. 81). We comprehensively determined the venom repertoires of the two *Anastatus* wasps by examining the contents (i.e., venom fluids) in the venom reservoir using mass spectrometry, together with RNA sequencing to estimate the expression level of candidate genes in the venom gland (see methods for details). In total, 257 and 210 venom genes were identified in *A. japonicus* and *A. fulloi*, respectively (Fig. 5b, c; Supplementary Data 16, 17). About half of the venom genes belong to the relatively young gene families, as they lack homologs in any other hymenopterans outside the superfamily Chalcidoidea (Fig. 5e).

We first studied the evident differences in venom composition between *Anastatus* and other 36 parasitoid wasps with available venom records. Venom gene content comparison showed that the most striking difference of venom gene copy number came from the serine protease family, which has 40 copies in *A. japonicus* and 41 in *A. fulloi*, whereas the most abundant one from other species only has 16 (Supplementary Data 19). In addition, we identified 18 *Anastatus*-specific venom gene families. These features could be relevant to the special parasitoid adaptation of *Anastatus*, e. g., parasitizing insect eggs, and thus our study also provides a valuable list of candidate genes for functional research.

We next focused on the divergent evolution of venom repertoires in the two *Anastatus* wasps by assigning orthologs among 19 hymenopterans. Our analysis revealed a total of 199 orthogroups (OGs) with venom genes, of which 43.7% (87) are strictly conserved (i. e., having the same copy number of venom genes within each OG) in the two *Anastatus*, which account for 103 genes in each species (Fig. 5d; Supplementary Data 19). Moreover, 52 *A. japonicus*-specific venom OGs and 25 *A. fulloi*-specific venom OGs were found.

To further unravel the venom evolution model in two *Anastatus*, we traced the origin of venom genes by analyzing the phylogenetic trees of venom OGs. Due to the potential for ambiguous inference, we first filtered out *Anastatus*-specific OGs and OGs with fewer than five gene members. 131 OGs were used for downstream analysis, representing about 72% (335) of *Anastatus* venom genes. Then, we screened the phylogenetic trees to pinpoint candidate genes for co-option venom recruitment and neofunctionalization followed by recent duplications. In terms of venom recruitment via co-option, we further detailed this model into three categories: (1) co-option of the universal single-copy genes, (2) co-option of genes formed by an ancient duplication outside the Chalcidoidea (about 105 mya), and (3) co-option of genes arose by a relatively recent duplication within Chalcidoidea, but prior to *Anastatus* divergence (about 3–105 mya) (See Supplementary Fig. 82 for the schematic diagram for each model). Our results showed that 14% of analyzed venom genes were evolved by co-option of universal single-copy genes, and an additional 12% were co-opted from the genes produced by an ancient duplication prior to Chalcidoidea divergence. Moreover, 22% were recruited from the genes with recent duplication inside the Chalcidoidea, but prior to *Anastatus* divergence (Supplementary Data 20). In total, the co-option model could explain approximately 48% of analyzed venom recruitments. On the other hand, 30% of analyzed venom genes were likely generated by neofunctionalization followed by recent duplications. Most (82%) of these neofunctionalizations had occurred in the common ancestor of the two *Anastatus* species and did not duplicate after their divergence. Overall, our findings concluded that the co-option and neofunctionalization models function concertedly to produce venom compositions in the two *Anastatus* species, with the co-option playing a predominant role during evolution, which is similar to the observations in other wasps[24]. The relatively high level of duplication-neofunctionalization and other recruitments in the *Anastatus*-specific OGs may be relevant to the higher gene family evolutionary rate of *Anastatus*. Additionally, we did not find strong evidence for rapid venom evolution at the sequence level, as only six venom genes show the significant signals of accelerated evolution, which was consistent with the previous finding in *Nasonia*[24] (FDR-adjusted $p < 0.05$, Supplementary Data 21).

Alternating gene expression is thought to be the primary mechanism of venom co-option evolution[24]. Thus, we assessed the expression patterns of venom genes across development through a comparative study in the two *Anastatus* species, involving total 202 one-to-one orthologous genes (Supplementary Data 22). Among them, 124 venom genes were shared between two species, implying ancestral venom recruitments. 55 genes possess venom functions in *A. japonicus*, while their orthologs in *A. fulloi* do not have venom functions. The remaining 23 genes are exclusively expressed in *A. fulloi*. Therefore, the last two gene sets represent recent venom changes after species divergence, providing us with a number of cases to understand the mechanism of rapid venom evolution. Generally, venom genes showed a strong tissue-specific expression profile, with high expression in venom gland and overall lower expression in the carcass (the rest tissues removing venom gland from an adult female wasp) and other developmental stages (Fig. 5f). The expression pattern of 124 shared venom genes was also similar to the general pattern (Fig. 5g). In contrast, the expression patterns of recently recruited venom genes and their non-venom orthologs were different, which showed much broader expression profiles across development (Fig. 5h, i). These expanded expression profiles of venom genes imply additional functions beyond venom. Correlation analyses of the expression level between venom genes and their non-venom orthologs revealed an overall high correlation (Spearman correlation coefficient over 0.9) in all developmental stages, but a lower correlation (Spearman correlation coefficient less than 0.4) in venom gland, suggesting that the large expression shift between these gene pairs occurred only in the venom gland (Fig. 5j; Supplementary Fig. 83), which thereby supports the mechanism of venom co-option (i.e., rapid expression evolution).

Intriguingly, we found several obvious non-coding rapidly evolving regions (NRERs) in the regulatory regions (upstream/downstream 1 Kb and introns) of genes with significant expression changes between the two species, which could explain their expression shifts. For example, *AKR1A1* (Aldo-keto reductase family 1 member A1) is highly expressed (TPM = 1417) in the venom gland of *A. japonicus*, but its ortholog in *A. fulloi* is barely expressed in the same tissue (TPM = 0.42). Whole genome alignment analysis showed a clear NRER (about 230 bp) in the 5′-proximal region of this gene, although the remaining sequences of this gene are well-aligned (Fig. 5k). Another example revealed that three NRERs in gene introns might also result in the expression changes, and notably, these NRERs are likely related to recent TE insertions after speciation (Fig. 5l). Comparative statistics

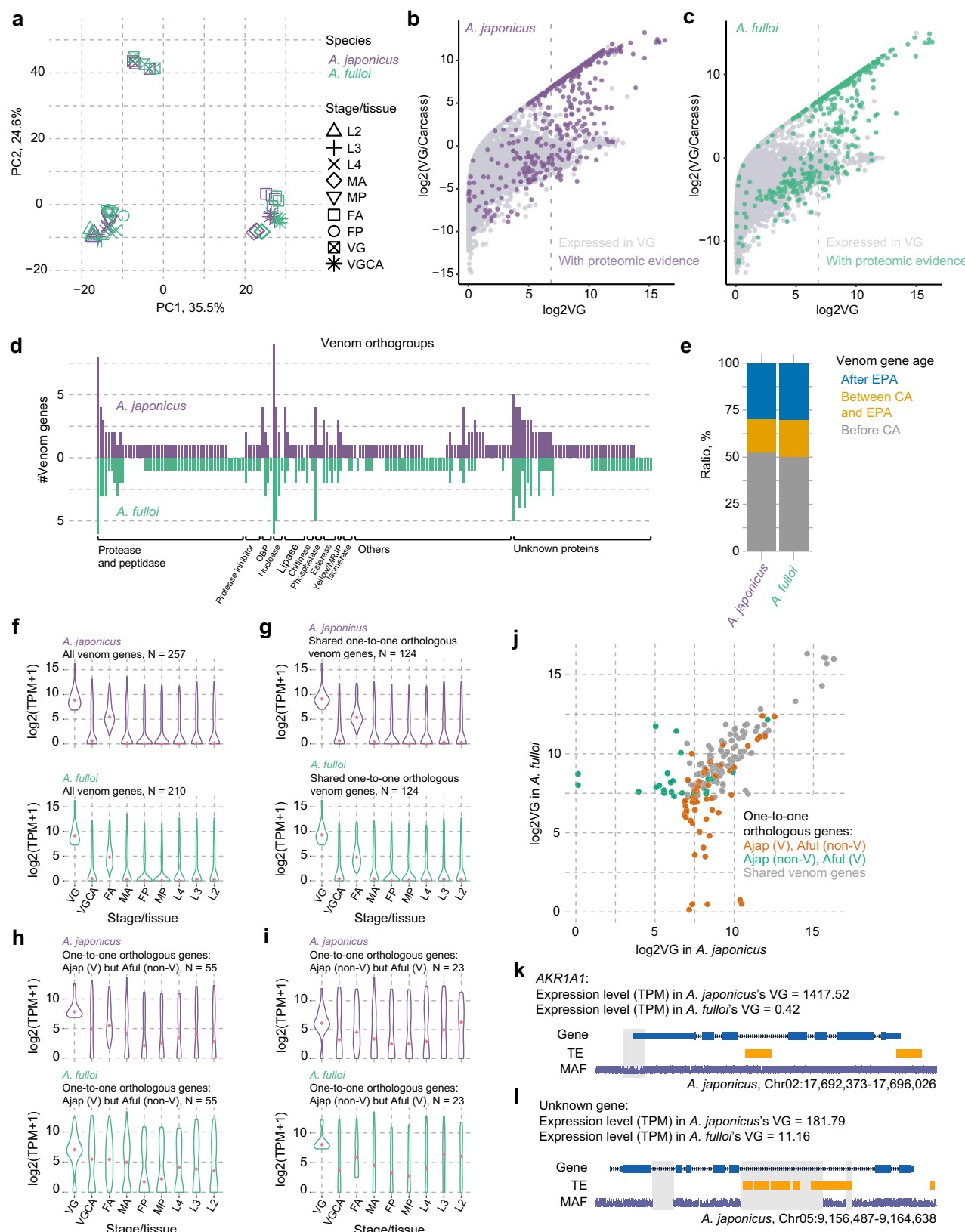

revealed that key regulatory regions related to the genes with large expression shifts (Foldchange > 6) between the two species are significantly enriched for NRERs when compared to the genes whose expression levels are relatively stable (Foldchange < 1.25) ($p = 0.001629$, chi-square test; Supplementary Table 20, Supplementary Data 23, Supplementary Fig. 84). Furthermore, recent TE

insertions explained 77.3% of the NRERs associated with the gene exhibiting expression shifts (Supplementary Data 23). Our observations imply that genomic variants, produced by recent TE insertions, may prompt gene expression shifts in the venom gland; however, further evidence is needed to test this hypothesis through expanded sampling and functional studies.

**Fig. 5 | Venom evolution of *Anastatus* wasps. a** Principal component analysis of gene expression in different developmental stages/tissues of the two *Anastatus* wasps. L2 2nd instar larva, L3 3rd instar larva, L4 4th instar larva, MP male pupa, FP female pupa, MA male adult, FA female adult, VG venom gland, VGCA carcass (i.e., adult female tissues without the venom gland). **b, c** Identification of venom genes in the two wasps. The gene expression level in the venom gland is presented on the *x*-axis, and the specialized level of expression in the venom gland is shown on the *y*-axis. Light dot, gene in the genome; Dark dot, gene supported by venom proteome. The vertical dashed line represents the genes with TPM values higher than the N90 value of the corresponding sample. **d** Comparison of venom orthogroups in the two wasps, See detail lists in Supplementary Data 19. **e** The age of venom orthogroups that estimated by the Wagner parsimony method. EPA, the ancestor of Eupelmidae and Pteromalidae; CA, the ancestor of superfamily Chalcidoidea.

**f** Expression of venom genes in the developmental stages/tissues of the two wasps. The expression pattern of 124 shared one-to-one orthologous venom genes in the two wasps (**g**), 55 one-to-one orthologous genes that are venom genes in *A. japonicus* but have lost venom functions in *A. fulloi* (**h**), and 23 one-to-one orthologous genes with venom functions in *A. fulloi* but have lost venom functions in *A. japonicus* (**i**) are also shown. **j** Dot plot showing the expression shifts in the venom gland of some one-to-one orthologous genes, which are highly relevant to the venom gene turnovers in the two wasps. **k** The NRER (gray region) in the 5´-proximal region of *AKR1A1* may be related to the expression differences in the venom gland between the two wasps. **l** NRERs in the intronic region of an unknown gene, which might also be related to the expression shift in the venom gland between the two wasps. And TE insertions contribute to the NRERs in this case. Source data are provided as a Source Data file.

## Venom gene coexpression network evolution in *Anastatus*

A gene coexpression network includes a group of genes with similar expression patterns. Because coexpressed genes are usually functionally connected, they are essential for understanding gene regulatory networks as well as protein-protein interactions[38]. However, the landscape of coexpression networks in parasitoids and how they evolve remain poorly understood. Here, we further constructed the venom gene coexpression networks of the two *Anastatus* species, and performed comparative analysis to trace the evolutionary path of the networks. Our weighted gene coexpression network analysis (WGCNA) constructed 38 and 37 modules in *A. japonicus* and *A. fulloi*, respectively (Supplementary Data 24, 25; Supplementary Fig. 85). As expected, in both species, the majority (81–86%) of identified venom genes were assigned to a single module (red module in *A. japonicus* and purple module in *A. fulloi*), which we named as venom-related network module (VRM). This module is absent from the network constructed using samples without venom gland, suggesting that VRM is highly venom gland specific (Supplementary Data 26, 27). The VRMs contained a total of 632 and 560 genes in *A. japonicus* and *A. fulloi*, respectively, and about 67% of them were non-venom genes (Fig. 6a, b). This observation implies that these non-venom genes coexpressed with venom genes may be relevant to venom production in parasitoid wasps. Following Gene Ontology enrichment analyses revealed that these genes were primarily involved in maintaining RNA location, glycoprotein catabolic process, protein localization, and response to unfolded protein (FDR-adjusted $p < 0.05$, Supplementary Data 28).

We next sought to investigate the divergence of VRM between the two *Anastatus* species. This analysis only employed unambiguous one-to-one orthologous gene pairs and focused on their module conservation and module shift. In the VRM of *A. japonicus*, 354 genes have clear orthologs in *A. fulloi*, and 59% (210) of them display module conservation between the two species. Similarly, 69% of genes in *A. fulloi*'s VRM we tested have their orthologs conserved in the VRM of *A. japonicus* (Supplementary Data 29, 30). These results reveal that extensive changes have occurred in VRMs after species divergence. In addition, we found that the majority (over 92%) of module shifts were significantly enriched in the non-venom gene set, while venom genes showed high module conservation between the two species ($p < 0.00001$ in both species, chi-square test; Supplementary Table 21). This finding suggests that module shifts have frequently happened to non-venom genes in VRM, which largely shaped the evolution of venom gene coexpression network.

Extensive changes in the venom gene coexpression networks between the two species prompted us to investigate whether they influence the core component of the network, making the network structure largely diverged. The core part of the network comprises a number of genes with high connectivity, also called hub genes, while the genes with relatively low connectivity are often located at the fringe area of a network. In this study, we employed module membership (also known as kME) to describe the connectivity of a gene with other module genes. The absolute values of kME range between 0

and 1, and a value close to 1 means a gene is highly connected to other genes in a module, whereas genes with a value closer to 0 represent low connectivity. Ortholog analysis of the non-venom genes in VRMs identified 549 OGs (Supplementary Data 31). Among them, only 96 (17.5%) OGs were shared between the two species and most of OGs were specific to a single species, further supporting our finding about the high turnover rate of non-venom genes in VRM. Genes belonging to shared OGs have significantly higher connectivity within the VRM when compared to those from the specific OGs (Fig. 6c; $p < 0.0001$, two-sided Wilcoxon rank-sum test). Moreover, our analysis revealed that the non-venom genes with high connectivity tended to stay in the VRM across species, while the lowly-connected genes were changeable (Fig. 6d). Venom genes also have overall higher connectivity when compared to non-venom genes (Supplementary Data 29, 30). Taken together, our results imply that although the VRMs have undergone extensive evolution, the core networks remained stable, whereas the major changes occurred in the periphery networks, which comprise many non-venom genes with relatively low network connections (Fig. 6e).

## Discussion

For a long time, scientists have been captivated by why genome sizes vary so greatly during the evolution of eukaryotes and how the genomes adapt to this variation[39]. In this study, the sequencing of two parasitoid wasps with much larger genomes (~950 Mb, 3–4 folds of most available parasitoid wasp genomes) presents a new opportunity to dissect the causes and consequences of genome size increase in Hymenoptera. Given the recently large-scale TE bursts occurred in the *Anastatus* genomes, we next asked whether *Anastatus* genomes had evolved an effective repression strategy to control TEs, as we identified intact TEs that may still be active and hence risk genomic stability. Here, we focused on a famous class of small RNAs usually derived from repetitive elements, piRNAs. In general, the Piwi-piRNA TE defense system mainly represses TEs and prevents their mobilization in the germline, which is a highly conserved mechanism in the Metazoa lineage, including mammals, birds, insects, etc.[16, 37, 40–43]. In addition, studies have demonstrated that somatic piRNAs (in contrast to germline piRNAs) also play a TE silencing role in the follicle cells[44], fat body[45], and brain[46] of *D. melanogaster*. An analysis of 20 arthropods, including Hymenopteran species, suggested that TE-targeting somatic piRNAs are common among arthropods[14]. In this study, our findings highlighted that the piRNA pathway was positively selected and enhanced as an adaptive response to target TEs in the *Anastatus* genomes. First, we found obvious gene family expansion of *Piwi* genes in the *Anastatus* genomes, and these *Piwi* genes exhibit a widespread expression pattern across almost all developmental stages and multiple adult tissues, implying that the piRNA pathway may provide comprehensive protection against TEs. Next, we sequenced small RNAs from both species to study whether piRNAs target TEs. We discovered abundant TE-derived piRNAs and significant Ping-Pong signals across TE families, which strongly supported the TE silencing role of

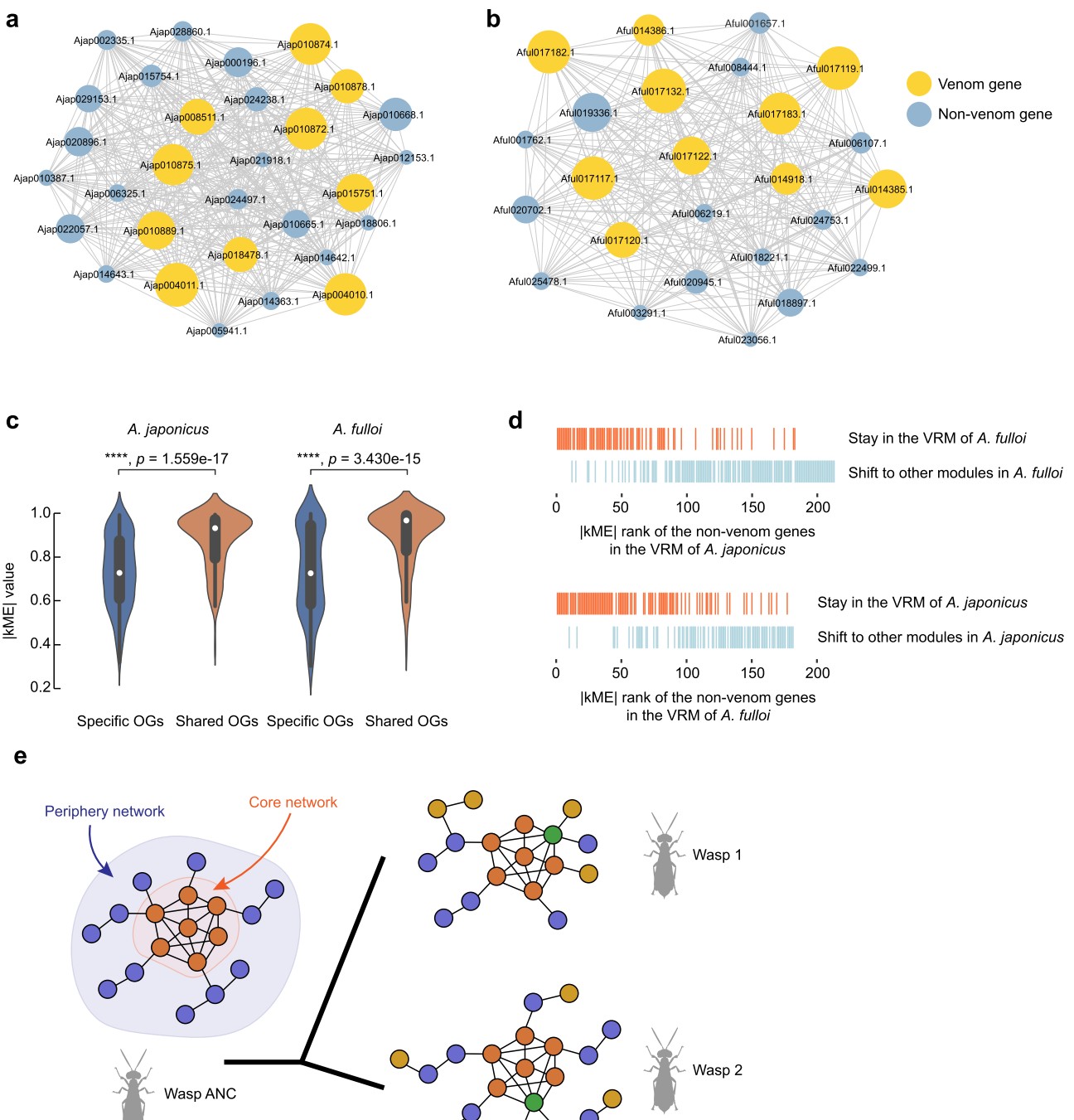

**Fig. 6 | The venom-related network module evolution in *Anastatus* wasps.** The venom-related network modules (VRMs) of *A. japonicus* (**a**) and *A. fulloi* (**b**), comprise a total of 632 and 560 genes, respectively. Since both networks have over 500 genes, visualizing the entire network topology would be impractical. Here, we only selected the top 20 highly expressed non-venom genes (in blue), and the top 10 highly expressed venom genes (in yellow) for visualization. The topic size indicates the expression level (log$_2$TPM). **c** The non-venom genes in VRM from the shared OGs between the two *Anastatus* wasps have significantly higher network connectivity than the non-venom genes from the specific OGs ($p < 0.0001$, two-sided Wilcoxon rank-sum test). $n_{\text{Ajap\_specific\_OGs}} = 297$; $n_{\text{Ajap\_specific\_OGs}} = 125$; $n_{\text{Aful\_speci}-}$ $_{\text{c\_OGs}} = 241$; $n_{\text{Aful\_specific\_OGs}} = 138$. Box plots displaying the interquartile range (IQR, the 25$^{\text{th}}$ and 75$^{\text{th}}$ percentiles) with median values (white dots), and whiskers extending to the highest and lowest points within 1.5× the IQR are shown over each violin plot.

**d** Module shift analysis of the one-to-one orthologous non-venom genes in VRM between the two *Anastatus* wasps. The rank of the |kME| value (network connectivity) of a gene in each wasp is presented on the *x*-axis. The rug plots above the *x*-axis show the rank distributions of genes in one wasp whose orthologs still keep in VRM (orange) or have shifted to other modules (light blue) in another wasp. **e** A schematic diagram for evolution of VRM in parasitoid wasps. A VRM could be roughly divided into two parts according to the network connectivity, i.e., core network and periphery network. During evolution, the periphery network shifts largely, which may be caused by network gene losses and new network gene recruitments (yellow points in Wasp 1 and Wasp 2). While the core network is more conservative with only relatively few changes (green points). Wasp ANC, wasp ancestor. Figure created with BioRender.com. Source data are provided as a Source Data file.

these piRNAs. Importantly, our results revealed that the features of TE-derived piRNAs, and their activities (Ping-Pong scores) are substantially linked with the TE characteristics of the *Anastatus* genomes. Taken together, the above TE defense strategies, which include the expansion of *Piwi* gene family, widespread expressions of *Piwi*, and active piRNA Ping-pong against TEs, may be explained by the red queen hypothesis, a well-known concept in evolutionary biology that describes the co-evolution between competing species (i. e., arms race). In insects, although the genome size and TE content vary largely during evolution[9, 13, 47], only limited studies focus on the arms race between TEs and piRNAs, and the majority of which were conducted in *Drosophila* species[48, 49], whose genomes are typically small (without obvious TE bursts). Therefore, our results provide a representative example of insects illustrating how the genome evolved in response to the recently exploded TEs.

We also found that the venom genes coexpressed with a large number of non-venom genes, forming a gene regulatory network (VRM). A similar gene regulatory network was previously described in the snake venom system[50]; however, the snake venom network (over 3000 genes) was much larger than that of the wasps (about 600 genes). By comparing the VRMs of two closely related wasps, we proposed that the changes of non-venom genes in the periphery part of the network largely shaped the evolution of VRM. We also speculated that these highly dynamic non-venom genes in VRMs could assist the evolution of venom compositions and functions in parasitoid wasps; however, their potential contributions have been unappreciated for a long time.

In summary, the two sequenced parasitoid wasp genomes, together with multi-omics data, advance our understanding of the causes and consequences of genome size evolution, as well as venom evolution. These genomic resources will also promote future comparative analyses of insects.

## Methods

### Genome and transcriptome sequencing

The genomic DNA of each species was extracted from about 50 haploid male pupae using the sodium dodecyl sulfate (SDS)-based DNA extraction method. To obtain the high-quality genomes, both PacBio Sequel II and Illumina Hiseq X Ten platforms were used for genome sequencing. For PacBio long-read sequencing, genomic DNA was used for 20 Kb SMRTbell library construction according to the manufacturer's protocol (Pacific Biosciences, CA, USA). HiFi reads enabled by circular consensus sequencing (CCS) of PacBio Sequel II system were used for constructing primary contigs. For short-read sequencing, the paired-end libraries (insert size of 300 bp) for each species were constructed and sequenced with the Illumina Hiseq Xten platform. We used the *k*-mer (17-mer) frequency-based method to estimate the genome size of each species using the jellyfish v2.2.10[51] and findGSE v1.94[52]. We further performed the Oxford Nanopore ultralong-read genome sequencing for assembly validation of each species. The genomic DNA of each species was size selected (>50 Kb) and processed for library construction using the Ligation sequencing 1D kit (SQK-LSK109, Oxford Nanopore). Each library was then sequenced on the Nanopore PromethION sequencer (Oxford Nanopore).

For transcriptome sequencing, total 27 samples from the 2nd instar larva, 3nd instar larva, 4th instar larva, female pupa, male pupa, female adult, male adult, venom gland, and carcass (whole body without venom gland) for each species were collected. Total RNA was extracted for all the samples using the TRIzol-based method (Invitrogen) and sequenced separately on the Illumina HiSeq X Ten platform with paired-end libraries (insert size of 300 bp). Three biological replicates were prepared for each sample. The raw RNA-seq reads were filtered by Fastp v0.20.0[53] and then were used for further RNA-seq analysis.

### Genome assembly and chromosome anchoring

First, 2,333,285 and 2,259,919 PacBio HiFi reads with a contig N50 = 14.7 Kb and 15.3 Kb were used for the primary genome assembly of *A. japonicus* and *A. fulloi*, respectively. The quality of these raw subreads was evaluated by the High-Quality Region Finder, which identifies the longest high-quality regain each read generated by a singly-loaded DNA polymerase according to the ratio of signal to noise. Then, CCS v6.0.0 was used to generate the consensus sequences with parameters:--min-passes 1 --min-rq 0.99 --min-length 100. We generated a draft assembly using Hifiasm v0.12[54] with parameter 'reads_cutoff:1k'. The primary assembly was polished with Illumina short reads for four rounds using Nextpolish v1.0.5[55]. Additionally, we performed Hi-C analysis to improve the primary genome assembly to the chromosomal level as we described in other genome projects[30, 31]. Hi-C libraries were prepared using freshly harvested haploid male pupae, and quantified and sequenced using the Illumina Novaseq platform (insert size is 150 bp). Clean Illumina pair-end reads were aligned to the contigs by bowtie2 v2.2.3[56] with parameters: --end-to-end --very-sensitive -L 30. Valid interaction paired reads were identified and filtered by HiC-Pro v3.1.0[57]. Finally, chromosome-level genomes were organized by LACHESIS[58]. The placement and orientation errors exhibiting obvious discrete chromatin interaction patterns were manually corrected. The final genome assembly of *A. japonicus* and *A. fulloi* comprises five chromosome-level scaffolds, and 94.69 % and 92.39% of the draft contigs were anchored and oriented successfully, respectively.

### Genome evaluation

The Benchmarking Universal Single-Copy Orthologs (BUSCO v5)[59] with insecta_odb10 was used to assess the genome assembly completeness. In addition, we also checked the completeness of the genome by mapping the short and long reads to the final assembly using BWA v0.7.17[60] and minimap2 v2.20[61] with default parameters, respectively. For HiFi and ONT reads, primary reads were filtered using Samtools v1.16[62] with parameter '-F 3844' and then used to calculate mean read depth across the assembly. Integrative Genomics Viewer (IGV) was used for visualization.

### Genome annotation

To identify TEs, we used the pipeline of extensive de novo TE annotator (EDTA)[63], which combines both structural-based and homology-based predictions. Briefly, species special TE libraries for each species were constructed using EDTA v1.9.9[63] with parameter '--sensitive 1'. Then, the transposable elements were identified using RepeatMasker v4.0.7[64] with the de novo library and Repbase library v16.2[65]. Full-length and solo LTR-RTs were identified using the LTR_FINDER_parallel[66] and LTR_retriever[67] implemented the EDTA pipeline. We also applied the same analysis to the other 17 hymenopteran genomes described in the Phylogenetic analysis section for further comparison.

To predict protein-coding genes, we adopted a strategy by integrating an ab initio gene prediction method, a homology-based gene prediction method and a transcriptional evidence-based method. First, the RNA-seq reads resulting from 27 samples of each species were aligned to the reference genome using HISAT2 v2.2.1[68]. Transcripts were then assembled using StringTie v2.1.0[69]. Next, we used Trans-Decoder v5.4.0 to predict complete coding sequences (CDS) of each transcript. For the homology-based gene prediction method, the protein sequences of Arthropoda from OrthoDB[70] were aligned to the repeat-masked genome by GenomeThreader v1.7.1[71]. For de novo gene prediction, BRAKER2[72] pipeline which integrates GeneMark-EP+[73] and AUGUSTUS v3.1[74] relying on the Arthropoda protein sequences was used. Finally, we used EVidenceModeler (EVM) v1.1.1[75] to integrate all evidence to produce the final high-confidence gene models. Functional annotation for these genes was performed by mapping their protein sequences to the following databases (SwissPro, Pfam, GO and

KEGG) using BLASTP or Hmmscan with default parameters (evalue < 1e−5). We also used BUSCO v5[59] with insecta_odb10 to assess the genome annotation completeness.

## LTR retrotransposon analysis

To estimate the insertion times of full length LTR retrotransposons, the paired LTR segments of each LTR retrotransposon were extracted and aligned by MAFFT v7.487[76] with default parameters. DISTMAT program implemented in the EMBOSS package[77] was then used to calculate the Kimura two-parameter distance (K) of each LTR pair. The insertion time (T) was calculated using the following formula: $T = K/2r$. The mutation rate (r) of each lineage used here was estimated by r8s v1.8.1[78] described below.

To construct the phylogenetic trees of LTR retrotransposons, the RT (reverse transcriptase) domains of each full-length LTR retrotransposons were identified by TEsorter v1.3.0[79]. Only the RT sequences without in-frame stop codons for each LTR retrotransposon were retained and aligned by MAFFT v7.487 with default parameters. After filtered by trimAl v1.2[80], the phylogenetic trees of *Copia*-type and *Gypsy*-type LTR retrotransposons were constructed using FastTree v2.1[81].

## Phylogenetic analysis

To infer the phylogenetic relationships of the *Anastatus* wasps and other hymenopterans, we selected additional 17 species across Hymenoptera for phylogenetic analysis. See Supplementary Table 22 for detailed information of these species. A total of 292,278 protein sequences from all 19 genomes were clustered into 20,983 orthologous groups using OrthoFinder v2.1[82] with parameters '-m MSA -T iqtree'. The protein sequences from 1792 single-copy orthologues were extracted from all 19 species and aligned by MAFFT v7.487[76] with L-INS-I model. After filtering by trimAl v1.2[80], these sequences were then concatenated to generate a supergene sequence, which was used to construct a maximum likelihood phylogenetic tree using IQ-TREE v2.0[83] with 1,000 replicates for ultrafast bootstrap analysis. The best-fitting model of sequence evolution (JTT + F + R6) estimated by ModelFinder[84] was used. The divergence times between species or clade were estimate by r8s v1.81[78]. Seven time points based on a previous study[29] were used to calibrate the tree: Orussoidea+Apocrita: 211–289 mya, Apocrita: 203–276 mya, Ichneumonoidea: 151–218 mya, Chalcidoidea: 105–159 mya, Aculeata: 160–224 mya, Apidae: 93–132 mya, and Formicidae: 65–127 mya.

## Gene family expansion and contraction

The gene family expansion and contraction were determined by CAFE v4.2.1[85] with the results from OrthoFinder and the phylogenetic tree with divergence times as inputs. Families with conditional *P* values lower than 0.05 were considered to have had a significantly accelerated rate of expansion or contraction.

## OR gene analysis

To annotate OR genes, the proteins sequences of previous reported insect ORs were first aligned to each genome sequence using Exonerate v2.4.0[86] with parameters: '--model protein2genome --maxintron --showtargetgff TRUE'. Then, InsectOR pipeline[87] was used to identify the ORs in each alignment region with parameters: '-tmh1 -tmh2 -tmh3 -p -m'. Only ORs with at least 300 amino acids in length were used in further analyses. The phylogenetic tree was constructed using the MAFFT-trimAl-IQ-TREE pipeline as described above.

## Argonaute gene superfamily and *Piwi* gene family expansion

To identify the genes from the Argonaute superfamily, well-curated insect Argonaute protein sequences were used as queries to search against the 19 genome sequences using tblastn (evalue 1e−5). Overlapping alignments were filtered and extended by 2000 bp upstream and downstream. The gene models in the alignment regions were predicted by Fgenesh+[88]. Each protein sequence obtained was subsequently used for searching against Pfam-A database by HMMscan v3.3.2[89] to identify the protein domains. Finally, each candidate Argonaute gene was manually inspected and divided into subfamilies. The phylogenetic tree was reconstructed by the MAFFT-trimAl-IQ-TREE pipeline. To further confirm the tandem duplications of *Piwi* genes, PacBio HiFi and ONT reads were mapped to the relevant genomic regions by minimap2 v2.20[61] and the coverages of all mapped reads and uniquely mapped reads were obtained by Samtools[62], and IGV was used for visualization. Codeml in Paml v4.9[90] was used to detect positive selection and calculate dN/dS ratios.

## Small RNA sequencing and piRNA analysis

Small RNA sequencing libraries were constructed from 50 adult females, for *A. japonicus* and *A. fulloi*, respectively. We selected small RNAs between 18–35 nt during the size selection step, to fully capture miRNAs, piRNAs, etc. An adapter was ligated to 3′ end of the single-stranded RNA. Then, the reverse transcription primer was hybridized to the 3′ adaptors and any excess 3′ adaptors. After ligation of the 5′ adaptors, the small RNA was reverse transcribed and followed by PCR amplification. Finally, the constructed small RNA libraries were sequenced using BGISEQ-500 platform.

To analyze the small RNA profiles, we first removed the 3′ adapter sequences using Trimmomatic v0.38[91]. Then we filtered out the reads that mapped to the following known sequences including rRNAs, tRNAs, snRNAs and snoRNA using bowtie v1.3.1[56] with '-k 1 -v 3' parameters. The rest reads were further used for miRNA prediction using miRDeep2 v0.1.3[92]. This prediction generated both miRNA precursor hairpin sequences and mature miRNA sequences. Small RNA reads were mapped to those predicted miRNA hairpin precursors using bowtie v1.3.1 with '-k 1 -v 3' parameters, and the unmapped reads were subsequently aligned to all known hairpin precursors in miRBase (Release 22.1) using the same parameters for evaluation, followed by removal from future analysis. For piRNA analysis, the rest reads with at least 24 nt were selected to be potential piRNAs. These reads were mapped to the genome using bowtie v1.3.1 with '-k 1 -v 1--best' parameters, which reported the best alignment, or if a read can be mapped to multiple locations, only one random location was chosen to report. The overlapping of these reads with different features including gene, and multiple types of transposons was performed by BEDtools v2.30.0[93]. On the other hand, these potential piRNA reads were mapped to the consensus sequences of transposon families, using bowtie v1.3.1 reporting all mapping locations, with at most two mismatches allowed. The 5′-to-5′ distance (from 0 to 30nt) of overlapping piRNAs mapped to the opposite strands of transposon consensus sequences was calculated as previously described[37]. The Z-score of the number of reads with 10 nt was calculated, which is also known as the Ping-Pong score. Finally, the small RNA reads were normalized to the genome mapping reads per million (RPM), and visualized using UCSC Genome Browser, separated by plus and minus strands of the genome. piRNA cluster analysis was performed using proTRAC[94] following their standard workflow using unique mapping reads. We first extracted the unique mapping reads from the genome mapping bam files using the "XM:i:1" tag. Then the reads were further filtered to remove low complexity sequences using TBr2_duster from the NGS TOOLBOX. These clean reads were further used for piRNA cluster prediction using the following parameters '-pdens 0.05 -1Tor10A 0.0 -clstrand 0.5 -distr 1-100'.

## qRT-PCR

We examined the expression levels of *Piwi* genes in different tissues in adult wasps. Briefly, total RNA was extracted by TRIzol Reagent (Invitrogen, Carlsbad, CA, US) according to the manufacturer's instructions. TransScript One-Step gDNA Removal and cDNA Synthesis

SuperMix Kit (TransGen Biotech, Beijing, China) were used for reverse transcriptions. qRT-PCR was performed using the Bio-Rad CFX 96 Real-Time Detection System (Bio-Rad, Hercules, CA, USA) with ChamQTM SYBR qPCR Master Mix Kit (Vazyme, Nanjing, China). The thermocycler was programmed for 95 °C for 3 min, 40 cycles of 95 °C for 10 s and 60 °C for 30 s. To verify the specificity of the amplification, a dissociation curve was included from 60 to 95 °C at the end of each qPCR run. The stably expressed 18 s rRNA gene was selected as reference gene, and the quantitative variation for each gene was calculated using a relative quantitative method ($2^{-\triangle\triangle CT}$). All primers used are listed in Supplementary Data 32.

### Transcriptome analysis
Raw reads of RNA-seq were filtered using Trimmomatic v0.38[91]. Transcripts were obtained by the HISAT2-StringTie pipeline we detailly described in the Genome annotation section. RSEM v1.3.3[95] was used for estimating the gene expression level with default parameters. Spearman's rank correlation coefficients were used to compare orthologues gene expression divergence among different samples.

### Venom proteome
Since the venom of parasitoid wasp is stored in venom reservoir until they're used, approximately 100 venom reservoirs of each species were isolated, pierced and washed for three times in sterile PBS. After centrifugation at 12,000 g for 10 min, the supernatant was collected and digested into peptides with trypsin. The digest peptides of each sample were desalted on C18 Cartridges (Sigma), and reconstituted in 40 μl of 0.1% (v/v) formic acid.

LC-MS/MS analysis was performed on a timsTOF Pro mass spectrometer (Bruker) that was coupled to Nanoelute (Bruker Daltonics). The peptides were loaded on a C18-reversed phase analytical column (homemade, 25 cm long, 75 μm inner diameter, 1.9 μm, C18) in buffer A (0.1% Formic acid) and separated with a linear gradient of buffer B (99.9% acetonitrile and 0.1% Formic acid) at a flow rate of 300 nl/min. The mass spectrometer was operated in positive ion mode. The mass spectrometer collected ion mobility MS spectra over a mass range of m/z 100–1700 and 1/k0 of 0.75 to 1.35, and then performed 10 cycles of PASEF MS/MS with a target intensity of 1.5k and a threshold of 2500. The active exclusion was enabled with a release time of 0.4 min. The raw data were processed using MaxQuant v2.0.3.1[96]. The protein sequences of each species were used as the database for searching.

### Identification of venom genes
Venom genes were identified by combining the transcriptomic and proteomic results. Briefly, the genes with TPM values higher than the N90 value of the venom gland were defined as highly reliable venom gland expressed genes. And a highly reliable venom gland expressed gene with at least three proteomic peptides which can be completely matched was qualified as a venom gene.

### Venom feature and evolution
To study the venom feature which is special in the *Anastatus* wasps, we used OrthoFinder v2.1 to compare the venom compositions among the two *Anastatus* and other 36 parasitoid wasps with venom records (from our venom database iVenomDB[97]). To investigate the venom divergence between the two *Anastatus*, we analyzed the difference of venom OGs (i. e., OG that comprise a venom gene of any of the two *Anastatus* wasps) based on the ortholog assignments with totally 17 hymenopteran species we obtained for comparative genomics and phylogenetics (See the Phylogenetic analysis section). In accordance with the parsimony principle, the age of each venom gene was assigned to different age groups based on the presence/absence of the orthologous genes in each species in the time tree[98].

To trace the evolutionary trajectory of venom genes, we constructed phylogenetic tree for each venom OG and manually check the evolutionary model for each venom gene. In brief, we first filtered the *Anastatus* specific OGs and the OGs with fewer than five gene members, which may lead to ambiguous inference. Then, protein sequences from each venom OG were extracted, aligned trimmed using MAFFT v7.487 and trimAl v1.2. The phylogenetic tree of each venom OG was built by IQ-TREE v2.0 with parameters: '-m MFP -B 1000'. We manually screened the phylogenetic trees to identify the evolutional origin and evolutionary model of each venom gene. Here, we tested for four venom evolutionary models: (1) co-option of the universal single copy genes, (2) co-option of the genes produced by an ancient duplication outside the Chalcidoidea, (3) co-option of the genes arose by a relatively recent duplication inside the Chalcidoidea, but before the *Anastatus* divergence, (4) venom recruitment and neofunctionalization followed by recent duplications. The schematic diagram for each co-option model was shown in Supplementary Fig. 82. The well-studied venom genes of *P. puparum* and *N. vitripennis* were marked in each phylogenetic tree of venom OG, which is helpful in locating the phylogenetic place of venom recruitments.

We performed whole genome alignment by LASTZ v1.04.03[99] with default parameters, to identify the interspecific structural variants and rapidly evolving regions. *A. japonicus* was used as reference. The alignments were visualized by UCSC Genome Browser. We manually checked the venom genes ($n = 202$) with unambiguous orthologues relationship between the two species, and searched for obvious NRERs (longer than 60 bp) in their potential key regulatory regions (upstream/downstream 1 Kb and introns). To test if the NRERs are significantly enriched in the key regulatory regions of genes with large expression changes (Foldchange > 6) in venom gland between the two species compared to the genes with relatively stable expressions between the two species (Foldchange < 1.25), we counted the number of NRERs in these two statuses respectively, and performed chi-square test. The causal relationship between NRER and recent TE insertion was determined by manually checking the UCSC Genome Browser with both TE track and alignment (MAF) track.

### Venom gene coexpression network
Weighted gene co-expression network analyses were constructed using the R package WGCNA v1.66[100]. Genes with TPM = 0 in all samples were filtered out. In total, for *A. japonicus*, 22,803 genes and 27 samples were used for coexpression network construction, and for *A. fulloi*, 24,191 genes and 27 samples were used. For WGCNA analyses, the detailed parameters that we applied as follows: '-network type = unsigned, -soft power = 4, -module identification method = dynamic tree cut, -minimum module size = 30, -the threshold to merge modules with a high similarity = 0.25'. To validate the tissue specificity of our constructed VRMs, we used the same parameters to re-run the WGCNA pipeline without venom gland in both two species, then evaluated the module preservation of previously identified VRMs in newly constructed results. Principal component analysis was performed on the genes in a module, and the most dominant component (PC1) was used to represent the traits of this module, i.e., the module eigengene (ME) vector. The module membership (MM), as known as kME, which is based on the Pearson correlation between gene expression values and ME, was used to describe gene connectivity to other module genes. The gene interaction networks were visualized using Cytoscape v3.8.2[101].

### Enrichment analysis
The python library GOATOOLS v1.0.6[102] was used for GO enrichment analyses.

### Reporting summary
Further information on research design is available in the Nature Research Reporting Summary linked to this article.

## Data availability

All sequencing data generated for this study are available from the National Genomics Data Center with accession number PRJCA008911.

The genome assembly data are available from the Genome Warehouse with accession number GWHBKAI00000000 and GWHBJYV00000000. The raw data of PacBio (CRA007569 and CRA007570), Illumina (CRA006534 and CRA006538), Hi-C (CRA007796 and CRA007800), ONT (CRA007785 and CRA007786), small RNA-seq (CRA007801 and CRA007802) and RNA-seq (CRA006651 and CRA006642) are available in the Genome Sequence Archive. Other public datasets used in this study include insecta_odb10 (https://busco-data.ezlab.org/v5/data/lineages/insecta_odb10.2020-09-10.tar.gz), all known miRNA hairpin precursors in miRbase (https://mirbase.org/ftp/CURRENT/hairpin.fa.gz), OrthoDB (https://v101.orthodb.org/download/odb10v1_all_fasta.tab.gz). Source data are provided with this paper.

## Code availability

All computational codes used in this study are available at https://github.com/yexinhai/Anastatus_genome_project[103] and archived at https://doi.org/10.5281/zenodo.7155373.

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

## Acknowledgements
This work was supported by Key Program of National Natural Science Foundation of China (NSFC) (Grant no. 31830074 to G.Y.Y.), the Program for Chinese Innovation Team in Key Areas of Science and Technology of Ministry of Science and Technology of the People's Republic of China (Grant no. 2016RA4008 to G.Y.Y.), the Program of NSFC (Grant no. 32001968 to C.Z.), the China Agriculture Research System of MOF and MARA (Grant no. CARS-32-12 to D.S.L), and the China Postdoctoral Science Foundation (Grant no. 2021M700125 to X.H.Y.).

## Author contributions
G.Y.Y., D.S.L. and X.H.Y. conceived and designed the study. F.W. supervised the work of X.H.Y. and provided feedback during development of the project. X.H.Y., Y.Y. and C.Z. led the manuscript preparation and writing with input from all co-authors, and G.Y.Y., D.S.L., Y.H.S., Q.F., S.X., S.J.X., X.X.Z. revised the manuscript. C.Z. performed the DNA extraction, and X.H.Y., Y.Y. performed the genome sequencing, assembly and annotation. C.Z. and Y.Y. prepared the samples for RNA-seq and small RNA-seq. X.H.Y. led all the analyses and experiments with assistances from co-authors. X.H.Y., Y.Y., C.H., B.Z., H.W.L. performed comparative genomics analyses and TE analyses. Y.H.S., X.H.Y., and S.X. carried out the piRNA analyses, *Piwi* gene analyses and related validations. X.H.Y., S.J.X., Y.Y. analyzed the venom evolution. X.H.Y. and Y.Y. performed the RNA-seq analyses. Q.F., Y.M., J.M.S. and H.X.X. participated in discussions. All authors read and approved the manuscript.

## Competing interests
The authors declare no competing interests.
