## [Peer Review File · Nature Communications]

REVIEWER COMMENTS

Reviewer #1 (Remarks to the Author):

The paper reports two good quality genomes (in itself this is now commonplace). These are large (although not exceptionally so), and the large size is associated with a high TE content (as is commonly observed in other species). There are several features of these genomes which were remarkable. First, there has been a recent and large acceleration in the rate of gene duplication. Second, there has been a massive and recent expansion of piwi genes, which are critical to silencing piRNAs. Third, the composition of venom proteins, which play a critical role in sabotaging the defences of parasitoid hosts changes extremely fast as new genes are recruited to venom function (this has been reported in other species, but this data is especially clear-cut). The text is clear and well-written.

The only substantive area I found weak was the suggestion that TE insertions were altering cis-regulatory elements, causing genes to switch to expression in the venom gland. This needs stronger evidence, for example examining the expression of reporter constructs. I suggest this claim is very clearly framed as speculation in all places and removed from the abstract.

The dramatic increase in gene number is very striking but deserves a little more analysis. Figure 1c: in the legend or main text it needs to be made clear what the numbers are. I am guessing the numbers refer to the terminal branches on the tree? This could do with correcting for branch length and measuring rates of gene birth and death across the tree (eg plot a phylogenetic reconstruction of the rate of gene gain and loss across the tree, colouring branches?). It is unclear from the current analysis if the increased rate is just gene gain or also gene loss. I would like to see robust evidence that this pattern is real and not caused by an assembly error where alleles of genes are assembled into separate contigs. This could likely be bioinformatic analysis (coverage duplicated vs non-duplicated?) rather than experimental. I note this is done to some extent for the piwi genes.

The massive expansion of the piwi family is perhaps the most remarkable result. Here I felt that Extended data figure 7 was important and poorly described in the text. The key result here is testis vs ovaries vs soma – it is clear there are piwis specialised on all of these cell types. I suggest this is clearly described in the text, and example plots showing this are moved to the main text (FISH would be even better, although I am not suggesting this as a revision). In this respect it was a shame that these different tissues were not sequenced for piRNAs.

Page 10/Figure 4h attempts to define a 'piRNA producing loci', which are often called 'piRNA clusters'. This analysis should be done using only uniquely mapped piRNAs, as otherwise it is not clear if this was the source of the short RNA. It would be very interesting to know if piRNAs mainly come from dispersed elements or clusters. It would also be interesting to know if there are viral piRNAs (assemble piRNAs and check for RDRP sequences).

Figure 1b – this does not convey any useful information. The Y axis and colors are not defined. The grey lines are impossible to see. Either delete or revise

Fig 3C taxon labels are not readable.

P12. The distinction of the 'co-option model' vs 'neofunctionalisation model' is unclear. In supplementary Figure 9 everything involves co-option. This needs clarifying.

A minor stylistic comment is that the authors might have gone further to move away from a generic 'genome paper' to focus on the interesting biology. This is intended as an observation, not a request for a revision.

Reviewer #2 (Remarks to the Author):

Review of NCOMMS-22-14803-T: Genomic signatures associated with maintenance of genome stability and venom turnover in two parasitoid wasps

In this study, the authors sequence and assemble reference genomes for two wasp species in the genus *Anastatus*. Using these new resources in a comparative genomics context, they analyze and report on aspects of genome structure, repetitive content, potential mechanistic interventions to combat transposable element proliferation, and venom gene evolution and regulation. They find that the two *Anastatus* genomes are among the largest hymenopteran genomes studied to date, and that larger genome size in these species is in large part due to expansion of LTR retrotransposons. They also report expanded tandem arrays of piwi genes, which, in concert with piRNA activity, may aid in suppressing TE proliferation to some degree. They further investigate venom genes and suites of co-expressed genes with putative functions in venom production, storage, etc.

The genome assemblies are of high quality and the data presented in the manuscript will be valuable for further comparative genomics studies. However, there are several major issues with the current manuscript that, if addressed, would provide additional confidence in the authors' conclusions, which I will detail below.

Major concerns:

1. A consistent concern throughout the manuscript is a lack of clarity/detail when reporting key findings. These are often presented in a very qualitative manner, lacking any apparent statistical test to support a given conclusion or to provide context for why it is being mentioned. For example:

Line 111: 'some structural variations were still identified' – How many? Where are they located? How were they determined? Any quantification to provide meaning here would be helpful.

Line 115: '944 are highly conserved across the two species' – what does highly conserved mean here?

Line 173: 'total length of Copia LTRs is much longer than' – How long, specifically? Is this supported using a statistical test?

There are numerous examples of these types of statements in the manuscript that should be revised to provide detailed, quantitative explanations of the authors' findings, including statistical tests, where appropriate.

2. There are also potential issues with the piwi expansion hypothesis and its relevance for suppressing TE activity in the genome. Namely, the authors indicate that piwi regions are free of misassembly errors on the basis of manual inspection of PacBio CCS reads – this is not a sufficient validation to support the absence of misassembly errors and is also not transparent/reproducible. The authors should provide a quantitative explanation of these results, perhaps using the distribution of reads overlapping multiple annotated piwi genes, for example. Further, they should quantify relative read depths in these regions, perhaps based on both long and short reads (if available), to verify that there is no bias in depth of uniquely-mapping reads compared to other genomic regions. They also generated Hi-C data, which should be used to show a lack of biased contact frequencies in these regions that would indicate misassembly errors.

3. Inferences of 'rapid' venom evolution seem to be primarily based on differences in inferred copy number or presence/absence of venom orthogroups between the two *Anastatus* species, rather than tests of evolutionary rates for orthologous sequences, which are typically used to test for rapid evolution at the sequence level. Tests of evolutionary rates using the phylogenetic trees already assembled for model testing of recruitment/cooption scenarios could be useful for determining which venom genes have experienced bursts of rapid evolution in *Anastatus*, specifically.

Further, it is possible that differences in gene copy number and presence/absence between the two species could be due to assembly artifacts. The authors could use reciprocal mapping experiments between the two species to verify that differential presence/absence of paralogs in one species is not due to assembly artifacts, as evidenced by reads mapping uniquely in one species and not the other.

Minor Comments:

Lines 104-108: The reporting BUSCO results seems inconsistent here. 97% complete BUSCO genes are reported, followed by 94% BUSCO completeness – is there a specific reason for the discrepancy?

Line 118: 'found massive TEs' – should this be 'found massive amounts of TEs'?

Lines 147-150: The expansion of olfactory receptors seems to be shared by *Nasonia* as well, if I'm interpreting the results correctly.

Line 206: It would be helpful to show genome size values for easy comparison between *Anastatus* and other wasps (perhaps in the phylogeny in Fig. 1?). Up until this point the manuscript reads as if the *Anastatus* genome size is unprecedented, yet *Belonocnema* has a much larger genome.

Response to reviewers:

Reviewer #1 (Remarks to the Author):

The paper reports two good quality genomes (in itself this is now commonplace). These are large (although not exceptionally so), and the large size is associated with a high TE content (as is commonly observed in other species). There are several features of these genomes which were remarkable. First, there has been a recent and large acceleration in the rate of gene duplication. Second, there has been a massive and recent expansion of piwi genes, which are critical to silencing piRNAs. Third, the composition of venom proteins, which play a critical role in sabotaging the defences of parasitoid hosts changes extremely fast as new genes are recruited to venom function (this has been reported in other species, but this data is especially clear-cut). The text is clear and well-written.

Response: We thank reviewer #1 for your helpful comments and suggestions. In response to the points raised by the reviewer, our current manuscript provides revised interpretation, additional analysis on gene gain/loss rates, and careful evaluation of the quality of genome assemblies, as you suggested.

The only substantive area I found weak was the suggestion that TE insertions were altering cis-regulatory elements, causing genes to switch to expression in the venom gland. This needs stronger evidence, for example examining the expression of reporter constructs. I suggest this claim is very clearly framed as speculation in all places and removed from the abstract.

Response: Thank you for the suggestion, and we share the same view on this. We have removed relevant statements from the abstract, introduction, and discussion sections. We also have weakened our conclusion and extended discussions followed by the observation of TE insertions: "Our observations imply that genomic variants, produced by recent TE insertions, may prompt gene expression shifts in the venom gland; however, further evidence is needed to test this hypothesis through expanded sampling and functional studies." (Line 426-428). In addition, the original figure about the proposed model for venom evolution mediated by TE insertions was removed from the Fig. 5.

The dramatic increase in gene number is very striking but deserves a little more analysis. Figure

1c: in the legend or main text it needs to be made clear what the numbers are. I am guessing the numbers refer to the terminal branches on the tree? This could do with correcting for branch length and measuring rates of gene birth and death across the tree (eg plot a phylogenetic reconstruction of the rate of gene gain and loss across the tree, colouring branches?). It is unclear from the current analysis if the increased rate is just gene gain or also gene loss. I would like to see robust evidence that this pattern is real and not caused by an assembly error where alleles of genes are assembled into separate contigs. This could likely be bioinformatic analysis (coverage duplicated vs non-duplicated?) rather than experimental. I note this is done to some extent for the piwi genes.

Response: Thank you for the suggestions. We confirm that the numbers in Fig. 1b refer to numbers of gene family expansion and contraction on the terminal branches and common ancestor branch of two *Anastatus* wasps. We have clearly indicated in the figure legend.

In response to your suggestions, we constructed the phylogenetic trees with colored branches to reflect gene gain/loss rates (Supplementary Fig S6, see below). Our analysis indicated that the two *Anastatus* wasps have not only a high rate of gene gain but also a high rate of gene loss, compared to other species in this study. We have included the following statement in the manuscript: “and they have high gene gain rates compared to the other species in our analysis (434 and 317 per My for *A. japonicus* and *A. fulloi*, Supplementary Fig S6, Supplementary Table 16). We also found relatively high gene loss rates of 51 and 56 per My for the terminal branches of these two wasps, respectively”. (Line 146-149)

Supplementary Fig. 56 | Rate of gene gain (a) and loss (b) along the hymenopteran phylogeny. All rates are color-indicated as branches of the phylogenetic tree. The phylogenetic tree was obtained from Fig. 1b.

We also thank the reviewer for pointing out the issues related to the quality of genome assemblies. To further validate the quality of these two chromosomal-level genomes, we first sequenced the genomic DNA using the Oxford Nanopore Technologies (ONT) platform for both species. More than 99% of the ONT reads can be mapped to assembly scaffolds, including over 30,000 reads longer than 100 Kb that were aligned uniquely and consistently. We also calculated the mapped sequencing read depth (PacBio HiFi and ONT) across the chromosome-scale scaffolds. Overall, we found uniform coverage across all chromosomes, with 99.9% of the assembly having coverages within three standard deviations of the mean values for either PacBio HiFi or ONT, which was used as a “gold standard” for sequencing coverage evaluation in the human T2T genome project (Nurk et al., 2022) (Supplementary Fig 4 for *A. japonicus*, see below). It was worth noting that we did see few regions with increased or decreased coverage, and there was only one region with increased coverage can be supported by both HiFi and ONT (for example, LG03:114,800,000-114,900,000). After carefully inspecting this region, we found a significant TE-rich pattern, and no protein-coding genes were found (Supplementary Fig 5).

Supplementary Fig. 4 | Sequencing coverage of *A. japonicus*. Chromosome-level genome coverage of mapped PacBio HiFi and ONT reads is shown with primary alignments.

Supplementary Fig. 5 | UCSC genome browser view of the region with increased coverage supported by PacBio HiFi and ONT in *A. japonicus* genome (LG03:114,800,000-114,900,000).

We also compared the sequencing coverage of duplicated genes identified by BUSCO, and we found the uniform sequencing coverage between the paralogs of duplicated BUSCO genes in the chromosome-scale scaffolds of two wasps (see Supplementary Fig 7 for example, and all relevant figures in Supplementary Fig 7–55).

Supplementary Fig. 7 | IGV browser view of PacBio HiFi and ONT reads mapped to the duplicated BUSCO gene (geneid: 22836at50557) in *A. japonicus*.

Based on the results of ONT ultra-long read mapping, sequencing coverage of the whole genome and duplicated gene regions, we confirmed the overall accuracy of the assemblies. And we have described the above results in Line 109-119.

Reference:

Nurk S, et al. 2022. The complete sequence of a human genome. Science. 376(6588):44-53.

The massive expansion of the piwi family is perhaps the most remarkable result. Here I felt that Extended data figure 7 was important and poorly described in the text. The key result here is testis vs ovaries vs soma – it is clear there are piwis specialised on all of these cell types. I suggest this is clearly described in the text, and example plots showing this are moved to the main text (FISH would be even better, although I am not suggesting this as a revision). In this respect it was a shame that these different tissues were not sequenced for piRNAs.

Response: We thank the reviewer for the suggestions. We have moved three representative examples showing broad *Piwi* expressions to the main Figure (Fig. 3). These three examples clearly showed three distinct patterns of *Piwi* gene expression, and we also have added corresponding statements to describe the findings: “Specifically, we measured *Piwi* expression on mRNA level in the reproductive organs (ovary and testis) and other somatic tissues. In *A. japonicus*, as expected, we observed the highest expression of all *Piwi* genes in the reproductive organs. However, in addition to finding *Piwi* genes that were specifically highly expressed in reproductive organs (e.g., *Piwi6* in Fig. 3c), we also found genes with significantly detectable expressions in the other analyzed tissues as well. For example, *Piwi2* was also highly expressed in the head compared to other tissues ($p < 0.05$, Tukey's multiple comparison test), reaching 45% of testis expression. Also, in relation to testis expression, *Piwi14* was expressed in diverse levels ranging from 12% (in digestive tract) to 68% (in ovary).” (Line 276-284). In addition, we totally agreed that studying piRNAs in different tissues including testis, ovaries, and soma, will be very interesting; however, we have no available data at hand, and we look forward to the follow-up studies in the future.

Page 10/Figure 4h attempts to define a ‘piRNA producing loci’, which are often called ‘piRNA clusters’. This analysis should be done using only uniquely mapped piRNAs, as otherwise it is not clear if this was the source of the short RNA. It would be very interesting to know if piRNAs mainly come from dispersed elements or clusters. It would also be interesting to know if there are viral piRNAs (assemble piRNAs and check for RDRP sequences).

Response: We thank the reviewer for the insightful suggestions. To identify the potential ‘piRNA clusters’ in *A. fulloi* and *A. japonicus*, we used proTRAC (Rosenkranz and Zischler, 2012),

a well-established piRNA cluster prediction tool to define piRNA clusters in their genome. First, we tested the software using public small RNA-seq data from *Drosophila* (Gainetdinov et al., 2018), and followed the standard piRNA cluster identification strategy using unique mapping reads that are over 23nt (see the Method section for detailed parameters). Our analysis yielded high sensitivity, which identified 97.4% of major piRNA clusters (111 out of 114) annotated in *Drosophila* (piRNA cluster annotated downloaded from <https://github.com/bowhan/piPipes/blob/master/common/dm6/dm6.piRNAcluster.bed.gz>). We further applied the same workflow and identified 1,111 (1.4% of its genome, see Supplementary table 34) piRNA clusters in *A. fulloi*, and 1,020 (1.3% of its genome, see Supplementary table 33) piRNA clusters in *A. japonicus*, respectively. We also updated Figure 4h, and additional Supplementary Fig 80, accordingly, showing typical examples of predicted piRNA clusters. Thus, we conclude that piRNAs in *A. fulloi* and *A. japonicus* are produced from piRNA clusters.

To further characterize whether piRNAs come from viruses, we assembled the piRNA sequences from *A. fulloi* and *A. japonicus* using Trinity, but failed to assemble any sequenced that can be attributed to RDRP using BLAST. To further investigate potential virus-derived piRNAs, we mapped piRNA reads to six known viruses (from Wang et al., 2021) in pteromalid wasps which are closely related to *A. fulloi* and *A. japonicus* (because we know little about the virus in *A. fulloi* and *A. japonicus*). As shown in the summary table below for viral piRNA mapping, very few reads can be mapped to these viruses when allowing maximum 3 mismatches using bowtie. Moreover, these viral mapping piRNAs can also be mapped to the genome. Although in mosquitos, piRNAs can be produced from viruses (Morazzani et al., 2012), we didn't observe solid evidence showing *A. fulloi* and *A. japonicus* share the same piRNA-producing strategy.

Summary Table of piRNA reads mapping to six known viruses

Virus length (nt)	Aful.Over23 (reads)	Ajap.Over23 (reads)	Aful.Over23 (species)	Ajap.Over23 (species)

Total					
reads/species	-	22,037,783	21,192,468	5,495,331	5,080,269
count					
AcNSRV-1	12,702	26	22	8	11
AcNSRV-2	12,118	39	3	8	3
AcPSRV-1	7,558	10	9	9	5
LdNSRV-1	11,633	4	4	4	4
WWPSRV-1	12,358	295	19	32	12
WWPSRV-2	9,419	3	13	3	4

Species: unique reads

Reference:

Gainetdinov I, Colpan C, Arif A, Cecchini K, Zamore PD. 2018. A single mechanism of biogenesis, initiated and directed by PIWI proteins, explains piRNA production in most Animals. *Molecular Cell*, 6;71(5):775-790.e5.

Morazzani EM, Wiley MR, Murreddu MG, Adelman ZN, Myles KM. 2012 Production of virus-derived Ping-Pong-dependent piRNA-like Small RNAs in the mosquito *Soma*. *PLoS Pathogens*, 8(1): e1002470.

Rosenkranz, D., Zischler, H. 2012. proTRAC - a software for probabilistic piRNA cluster detection, visualization and analysis. *BMC Bioinformatics*,13, 5.

Wang et al., 2021. Diverse RNA viruses discovered in three parasitoid wasps of the rice Weevil *Sitophilus oryzae*. *mSphere*, 6, (3):e00331-21.

Figure 1b – this does not convey any useful information. The Y axis and colors are not defined.

The grey lines are impossible to see. Either delete or revise

Response: We have removed these results from the manuscript.

Fig 3C taxon labels are not readable.

Response: We have revised Fig. 3 to make the labels easier to read.

P12. The distinction of the ‘co-option model’ vs ‘neofunctionalisation model’ is unclear. In

supplementary Figure 9 everything involves co-option. This needs clarifying.

Response: Thank you for this comment. We have revised the Supplementary Fig 82, to articulate the two evolutionary models.

A minor stylistic comment is that the authors might have gone further to move away from a generic 'genome paper' to focus on the interesting biology. This is intended as an observation, not a request for a revision.

Response: We thank the reviewer for this observation. We agree entirely with the reviewer that we would like to make this paper to focus on the interesting biology, rather than presenting it as a traditional data article. We set our goal beyond a general 'genome paper' and demonstrate how high-quality genomes power the understanding of fundamental biological questions, which we believe, aligns well with the top-tier research published in *Nature Communications*.

Reviewer #2 (Remarks to the Author):

Review of NCOMMS-22-14803-T: Genomic signatures associated with maintenance of genome stability and venom turnover in two parasitoid wasps

In this study, the authors sequence and assemble reference genomes for two wasp species in the genus *Anastatus*. Using these new resources in a comparative genomics context, they analyze and report on aspects of genome structure, repetitive content, potential mechanistic interventions to combat transposable element proliferation, and venom gene evolution and regulation. They find that the two *Anastatus* genomes are among the largest hymenopteran genomes studied to date, and that larger genome size in these species is in large part due to expansion of LTR retrotransposons. They also report expanded tandem arrays of *piwi* genes, which, in concert with piRNA activity, may aid in suppressing TE proliferation to some degree. They further investigate venom genes and suites of co-expressed genes with putative functions in venom production, storage, etc.

The genome assemblies are of high quality and the data presented in the manuscript will be

valuable for further comparative genomics studies. However, there are several major issues with the current manuscript that, if addressed, would provide additional confidence in the authors' conclusions, which I will detail below.

Response: We thank Reviewer #2 for his/her time in reviewing our work. And especially, we appreciate his/her insightful, constructive suggestions, which are critical to improving our manuscript.

Major concerns:

1. A consistent concern throughout the manuscript is a lack of clarity/detail when reporting key findings. These are often presented in a very qualitative manner, lacking any apparent statistical test to support a given conclusion or to provide context for why it is being mentioned. For

example:

Line 111: 'some structural variations were still identified' – How many? Where are they located?

How were they determined? Any quantification to provide meaning here would be helpful.

Line 115: '944 are highly conserved across the two species' – what does highly conserved mean here?

Line 173: 'total length of Copia LTRs is much longer than' – How long, specifically? Is this supported using a statistical test?

There are numerous examples of these types of statements in the manuscript that should be revised to provide detailed, quantitative explanations of the authors' findings, including statistical tests, where appropriate.

Response: Thank you for this critical comment. We have gone through the manuscript and added detailed and quantitative explanations for many findings. Necessary statistical tests have been provided. Additional Supplementary Tables/Figures, and Source data were also included to support the statements.

2. There are also potential issues with the piwi expansion hypothesis and its relevance for suppressing TE activity in the genome. Namely, the authors indicate that piwi regions are free of misassembly errors on the basis of manual inspection of PacBio CCS reads – this is not a sufficient validation to support the absence of misassembly errors and is also not

transparent/reproducible. The authors should provide a quantitative explanation of these results, perhaps using the distribution of reads overlapping multiple annotated *piwi* genes, for example. Further, they should quantify relative read depths in these regions, perhaps based on both long and short reads (if available), to verify that there is no bias in depth of uniquely-mapping reads compared to other genomic regions. They also generated Hi-C data, which should be used to show a lack of biased contact frequencies in these regions that would indicate misassembly errors.

Response: Agree. Thank you for the suggestion. In the revision, we systematically check the *Piwi* gene regions by inspecting long reads (including original PacBio HiFi reads and our newly sequenced ONT ultra-long reads) and calculating the mapped sequencing read depths. We provided IGV screenshots for each *Piwi* gene and the flanking regions (Supplementary Fig. 60-76), together with multiple Supplementary Tables (Supplementary Table 24-27) to summarize the sequencing coverage (including the gene region, upstream 10k region, and downstream 10k region). In addition, taking advantage of the ultra-long ONT reads, we identified all ONT reads spanning multiple *piwi* genes, serving as extra evidence of *Piwi* duplication (results available in Supplementary Table 28-29). Based on these analyses, we have evaluated and quantified multiple metrics to verify the genome quality, and in our opinion, the data processing and validation procedures are now transparent and reproducible. We did not use the Hi-C data to validate the accuracy of genome assembly because Hi-C was used for scaffolding the genome to chromosome level, and our data only supported a relatively low resolution which may be inappropriate for detecting contact frequencies in small-scale regions.

In general, we observed uniform read coverage across *Piwi* gene regions and their flanking regions, regardless of the mapping parameters we chose, such as default setting allowing multi-mappers or unique mapping only setting (see Supplementary Fig. 60 for example).

Supplementary Fig. 60 | PacBio HiFi and ONT reads spanning *Piwi* genes (*Piwi1*, *Piwi2*, *Piwi3*) in *A. japonicus*.

In addition, we did find coverage decreases in the regions with tandemly arrayed *Piwi* genes. However, we identified uniquely mapped ONT ultra-long reads, which can support all the intact gene arrays. For example, as shown in Supplementary Fig. 65 below, although the sequencing coverage (particularly for unique mapping) decreases a lot compared to other genomic regions, we found an ONT ultra-long read (readID: d863abd3-b92d-45a1-87f9-aa6b4c6e0c31, Supplementary Table. 28) which fully supports the whole *Piwi* gene cluster (*Piwi10-18*), providing strong evidence for the existence of this *Piwi* gene cluster. Given the uniformity of coverage across this region, association with largely duplicated *Piwi* genes, and the occasionally discrepant effect observed between PacBio HiFi and ONT, we hypothesize that these anomalies may be related to biases introduced during sample preparation or sequencing, rather than assembly error. We also found numerous sequences of *Piwi* on some short contigs with few other genes, suggesting that the repetitive nature of this region prevented these contigs from being assembled into chromosome-scale scaffolds, or, that they could be redundant artifacts. Given this uncertain status, the *Piwi* genes from these contigs were removed in downstream analyses. Based on the above information, we revised the

relevant statements about *Piwi* genes and downstream analyses. (Line 245-258)

Supplementary Fig. 65 | PacBio HiFi and ONT reads spanning the *Piwi* genes (*Piwi9-18*) in *A. japonicus*. The green box represents the gap (100Ns) concatenated contigs to create five chromosome-scale scaffolds.

Moreover, we recently noticed that the genome of another *Anastatus* wasp (*Anastatus disparis*) was available (GenBank assembly accession: GCA_017163975.1), but no gene annotations were accessible. To further validate the finding of largely *Piwi* duplication in *Anastatus* wasps, we manually annotated *Piwi* genes by TBLASTN and Fgenesh+, and found 40 *Piwi* genes in the contig-level genome of *Anastatus disparis*. This finding therefore strengthened our conclusion about *Piwi* duplication in *Anastatus* wasps. Here, considering that the genome submitter (or the person who sequenced this genome) did not publish this data publicly, we did not include this analysis in the revised manuscript.

3. Inferences of ‘rapid’ venom evolution seem to be primarily based on differences in inferred copy number or presence/absence of venom orthogroups between the two *Anastatus* species, rather than tests of evolutionary rates for orthologous sequences, which are typically used to test for rapid evolution at the sequence level. Tests of evolutionary rates using the phylogenetic trees already assembled for model testing of recruitment/cooption scenarios could be useful for

determining which venom genes have experienced bursts of rapid evolution in *Anastatus*, specifically.

Response: Thank you for this comment. We then tested the rapid venom evolution at the sequence level, using the branch model in Codeml software. The results showed only six venom genes that evolved rapidly compared to other non-venom genes (FDR-adjusted $p < 0.05$). We found that this observation was consistent with the previous finding of venom study in *Nasonia* species (Martinson et al., 2017). Therefore, we have added this sentence about sequence-level evolution: “Additionally, we did not find strong evidence for rapid venom evolution at the sequence level, as only six venom genes show the significant signals of accelerated evolution, which was consistent with the previous finding in *Nasonia*²⁴ (FDR-adjusted $p < 0.05$, Supplementary Table 40)” (Line 387-389).

Reference:

Martinson EO, Mrinalini, Kelkar YD, Chang CH, Werren JH. 2017. The evolution of venom by co-option of single-copy genes. *Current Biology*, 27(13):2007-2013.e8.

Further, it is possible that differences in gene copy number and presence/absence between the two species could be due to assembly artifacts. The authors could use reciprocal mapping experiments between the two species to verify that differential presence/absence of paralogs in one species is not due to assembly artifacts, as evidenced by reads mapping uniquely in one species and not the other.

Response: We appreciate this suggestion by the reviewer. In this study, we identified venom genes by considering the RNA expression in the venom gland and the proteomic evidence from venom. The results showed the large differences in copy number or presence/absence of venom orthogroups between the two *Anastatus* species, which were largely caused by the gene expression shifts in the venom gland or presence/absence of proteomic evidence, rather than the copy number variations or presence/absence of the orthogroups containing venom genes between the two *Anastatus* species. In fact, 63% (125/199) of these orthogroups have the exact the same gene copy numbers between the two *Anastatus* species, and the gene copy numbers in 92% (184/199) of these orthogroups varied within three. Based on these

explanations and further evaluation of genome assembly in this revision (original PacBio HiFi reads and newly sequenced ONT reads coverage across the whole genomes, read coverage of duplicated BUSCOs, Line 109-119), we concluded that the obvious assembly errors or artifacts were not in these two wasp genomes. In addition, we applied the methods suggested by the reviewer and performed the reciprocal mapping analysis between these two species. We found that only about 50% of short reads could be successfully cross-mapped, which may be due to the species divergence (they had diverged 3 million years), resulting in evident genomic differences that the aligner cannot accommodate. We thus concluded that this method is likely applied to population analysis or pan-genome projects to identify the presence/absence of variations, but may be inappropriate for our data.

Minor Comments:

Lines 104-108: The reporting BUSCO results seems inconsistent here. 97% complete BUSCO genes are reported, followed by 94% BUSCO completeness – is there a specific reason for the discrepancy?

Response: Thank you for pointing out this ambiguity. Actually, we did two independent BUSCO analyses for each species: (1) running BUSCO on genome assembly to evaluate the completeness of genome assembly, and (2) running BUSCO on genome annotation to evaluate the quality of the results of the automatic gene annotation pipeline. Here, the 97% BUSCO score was for genome assembly, and genome annotation BUSCO score was 94%. We have revised this statement: “A total of 27,792 and 27,168 protein-coding genes were annotated in *A. japonicus* and *A. fulloi*, respectively, and both of them were supported by over 94% BUSCO completeness, suggesting the comprehensive gene annotation (Supplementary Table 7).” (Line 106-109)

Line 118: ‘found massive TEs’ – should this be ‘found massive amounts of TEs’?

Response: Revised.

Lines 147-150: The expansion of olfactory receptors seems to be shared by *Nasonia* as well, if I’m interpreting the results correctly.

Response: Corrected. And it now reads: “Moreover, we found a larger number of olfactory receptor genes in both *Anastatus* (104 and 102) and *Nasonia* (72) genomes compared to honey bee (22), with a highly duplicated 9-exon subfamily”. (Line 157-159)

Line 206: It would be helpful to show genome size values for easy comparison between *Anastatus* and other wasps (perhaps in the phylogeny in Fig. 1?). Up until this point the manuscript reads as if the *Anastatus* genome size is unprecedented, yet *Belonocnema* has a much larger genome.

Response: Thank you for this suggestion. We have revised the Fig. 1b, to show the genome size for each species.

REVIEWERS' COMMENTS

Reviewer #1 (Remarks to the Author):

The authors have taken account of my comments and added significant new data and analysis. This addresses all my concerns and I have no new revisions to suggest. I consider this an important and fascinating paper.

Reviewer #2 (Remarks to the Author):

The authors have addressed my concerns in their revised manuscript. In particular, I appreciate their attention to clarify multiple points of confusion based on presentation of the original results, including new details on statistical analyses, etc. They have also sufficiently addressed suggested changes/additions to verify the robustness of their findings.